# Scaled, high fidelity electrophysiological, morphological, and transcriptomic cell characterization

Brian R Lee[1]*[†], Agata Budzillo[1]*[†], Kristen Hadley[1], Jeremy A Miller[1], Tim Jarsky[1], Katherine Baker[1], DiJon Hill[1], Lisa Kim[1], Rusty Mann[1], Lindsay Ng[1], Aaron Oldre[1], Ram Rajanbabu[1], Jessica Trinh[1], Sara Vargas[1], Thomas Braun[2], Rachel A Dalley[1], Nathan W Gouwens[1], Brian E Kalmbach[1,3], Tae Kyung Kim[1], Kimberly A Smith[1], Gilberto Soler-Llavina[1], Staci Sorensen[1], Bosiljka Tasic[1], Jonathan T Ting[1,3], Ed Lein[1], Hongkui Zeng[1], Gabe J Murphy[1,3], Jim Berg[1]

[1]Allen Institute for Brain Science, Seattle, United States; [2]Byte Physics, Berlin, Germany; [3]Department of Physiology and Biophysics, University of Washington, Seattle, United States

**\*For correspondence:**
brianle@alleninstitute.org (BRL);
agatab@alleninstitute.org (AB)

[†]These authors contributed equally to this work

**Competing interest:** The authors declare that no competing interests exist.

**Abstract** The Patch-seq approach is a powerful variation of the patch-clamp technique that allows for the combined electrophysiological, morphological, and transcriptomic characterization of individual neurons. To generate Patch-seq datasets at scale, we identified and refined key factors that contribute to the efficient collection of high-quality data. We developed patch-clamp electrophysiology software with analysis functions specifically designed to automate acquisition with online quality control. We recognized the importance of extracting the nucleus for transcriptomic success and maximizing membrane integrity during nucleus extraction for morphology success. The protocol is generalizable to different species and brain regions, as demonstrated by capturing multimodal data from human and macaque brain slices. The protocol, analysis and acquisition software are compiled at https://githubcom/AllenInstitute/patchseqtools. This resource can be used by individual labs to generate data across diverse mammalian species and that is compatible with large publicly available Patch-seq datasets.

## Introduction

Describing and understanding the properties of neuronal cell types is a critical first step toward understanding circuit activity within the brain, and ultimately cognitive function. Neurons exhibit stereotyped yet diverse electrophysiological, morphological, and transcriptomic properties (*Tasic et al., 2018*; *Tasic et al., 2016*; *Zeng and Sanes, 2017*; *Arendt et al., 2016*; *Kepecs and Fishell, 2014*; *Tremblay et al., 2016*; *Gouwens et al., 2020*; *Gouwens et al., 2019*) and understanding how each of these distinct features relate to one another may provide us with mechanistic insight into the roles of these neuron types. The introduction of single-cell RNA-sequencing (scRNA-seq) has revolutionized the field of transcriptomics (*Zeisel et al., 2015*). Dissociated cells or nuclei are isolated in a high-throughput manner to provide a comprehensive analysis of the molecular underpinnings of a single cell. Systematic and large-scale scRNA-seq approaches have been successful at characterizing brain cell types across mammalian species (*Yao et al., 2021*; *Tasic et al., 2016*; *Tasic et al., 2018*; *Bakken et al., 2018*; *Hodge et al., 2019*; *Hashikawa et al., 2020*; *Zeisel et al., 2015*; *Mickelsen et al., 2019*, *Zeisel et al., 2015*). These large-scale studies often include data from tens of thousands to millions of neurons, whereas electrophysiological or morphological studies are limited to tens or hundreds of neurons. Despite scRNA-seq providing an in-depth look into gene expression and cell

type classification, relationships to the morpho-electric neuron types described in the literature (*Tremblay et al., 2016*) can only be inferred. Studies with triple modality data are rare and lack the scale to capture the true biological variability.

The Patch-seq recording technique is a powerful approach (*Cadwell et al., 2017*; *Cadwell et al., 2015*; *Fuzik et al., 2015*; *van den Hurk et al., 2018*) that can provide morphology (M), electrophysiology (E), and transcriptome (T) data from single neuron, that is triple modality MET data. These data are an important tool toward establishing 'correspondence' of these properties through analysis. This technique is a modification of the well-established slice patch-clamp electrophysiology approach, where intrinsic properties are recorded from neurons in acute brain slices while simultaneously filling the neuron with biocytin. Following fixation, the biocytin is reacted with DAB as chromogen to generate a dark precipitate that fills the neuron, enabling imaging and a digital morphological reconstruction. In the Patch-seq technique, the neuron's cytoplasm is collected at the end of the recording, then processed via RNA-seq to identify its gene expression patterns. The technique has been used successfully to characterize both cortical and subcortical neurons (*Berg et al., 2020*; *Dudok et al., 2021*; *Scala et al., 2020*; *Hashikawa et al., 2020*; *Gouwens et al., 2020*; *Muñoz-Manchado et al., 2018*).

Despite its promise, early Patch-seq data suffered from three primary issues: (1) inconsistent quality, with non-specific cellular contamination and low yield of genes critical to accurate mapping to transcriptomic types (*Tripathy et al., 2018*), (2) little to no recovery of neuronal morphology (*Cadwell et al., 2015*), and (3) low throughput, with adequate cell fills requiring up to an hour of recording per cell. To address these issues, we refined the Patch-seq technique to increase the efficiency of each step to minimize data attrition and increase the throughput to facilitate a more comprehensive analysis of cell type characterization (*Gouwens et al., 2020*; *Berg et al., 2020*). In a large-scale manner, we systematically modified the existing Patch-seq protocols, using feedback from experimental metadata to reveal the key determinants of success, including nucleus extraction and slow withdrawal of the recording electrode (*Lipovsek et al., 2020*; *Cadwell et al., 2015*; *Fuzik et al., 2015*; *Cadwell et al., 2017*). Using a customizable electrophysiology acquisition package, we created specialized adaptive stimulus sets and online quality control to reduce experiment duration and increase throughput. Together with these adaptations, we demonstrate how high-quality triple modality information may be gathered across diverse neuronal types, different species and at scale.

Adopting the Patch-seq technique in an existing or new laboratory can be daunting due to the complexity of multiple data streams. To make adoption as simple as possible, we have created a resource, https://github.com/AllenInstitute/patchseqtools (copy archived at swh:1:rev:d1afcd-4d5203564979a29f2891e03cba7733b726; *Miller et al., 2021*), as a starting point for labs interested in using the Patch-seq technique or refining their existing technique. Specifically, this resource consists of three components: (1) a step-by-step optimized Patch-seq protocol, including helpful tips and a troubleshooting guide; (2) the Multichannel Igor Electrophysiology Suite (MIES) software package, including built-in analysis functions to increase the efficiency and robustness of Patch-seq data; and (3) an R library that uses a modified workflow from the patchSeqQC R library (*Tripathy et al., 2018*) to process and assay the quality of Patch-seq transcriptomic data. Here, we highlight the components of this resource and detail the background behind critical protocol decisions. The resource balances detailed internal standards with flexibility to adjust to a specific user's experimental approach. In the end, data generated using this resource can be benchmarked against the data from the over 7000 publicly available Patch-seq neuron experiments that can be downloaded from https://celltypes.brainmap.org/.

## Results

### A Patch-seq protocol optimized for fast, high-quality data generation

We used broad and specific transgenic Cre driver mouse lines to target over 8000 excitatory and inhibitory neurons in adult mouse primary visual cortex (VISp). Of these neurons, 4309 were previously included in a study (*Gouwens et al., 2020*) to characterize the morphological, electrophysiological, and transcriptomic properties of interneurons from the mouse visual cortex (*Figure 1—figure supplement 1*). The expanded dataset used for the analysis presented here includes cells that did not pass the stringent quality control (QC) metrics required to be part of that previous study. This is intentional,

as it allows us to evaluate and demonstrate the utility of these QC metrics, and their predictors, as well as protocol modifications that can maximize success. In addition, the current study includes excitatory neurons to demonstrate the generalizability of the metrics and protocol to different neocortical cell types.

To compare neurons within similar neuron types, we focused on four established Cre lines: *retinol binding protein 4* (Rbp4)-Cre for glutamatergic (excitatory) neurons and *parvalbumin* (Pvalb)-Cre, *somatostatin* (Sst)-Cre, and *vasoactive intestinal peptide* (Vip)-Cre for GABAergic (inhibitory) neurons. Each of these lines have been shown to be cell type- and/or brain region-specific (*Madisen et al., 2009*; *Harris et al., 2014*), and they label neurons within the same transcriptomic class (e.g. glutamatergic, GABAergic) or subclass (e.g. Pvalb, Sst, Vip) (*Tasic et al., 2016*; *Tasic et al., 2018*). Neurons within the same class or subclass exhibit similar morphoelectric properties (*Tremblay et al., 2016*; *Gouwens et al., 2019*; *Gouwens et al., 2020*), which should minimize biological variation and allow for a more appropriate technical comparison. These Cre lines are also widely used and publicly available, making them an ideal benchmark for troubleshooting for an adopting laboratory.

To evaluate success at each stage of the Patch-seq experiment, we defined a series of qualitative and quantitative parameters for each data modality (*Supplementary file 1*). Although we provide internal thresholds for each metric, they each exist along a continuum and could be differentially applied depending on the circumstances of each individual user. Given these criteria, the established protocol described here has a pass rate of 97%, 93%, and 46% for electrophysiology, transcriptomics, and morphology, respectively (*Figure 1*), with a final rate of successful triple modality, MET, data collection of 39 %. With morphological recovery having the highest rate of attrition, Patch-seq neurons that fail at this point but pass electrophysiology and transcriptomics (91 % in total) can still be used for spatial/anatomical, electrophysiological, and transcriptomic characterization and classification.

The patch-clamp portion of the protocol consists of three phases: recording (1–15 min), nucleus retrieval (< 3 min), and nucleus extraction (< 8 min) (*Figure 2*). The recording phase consists of whole-cell electrophysiology to acquire intrinsic features using custom, free, publicly available software: MIES (*Video 1* and *Video 2*). The recording period is kept as short as possible to increase throughput and reduce the effect of progressive cell swelling (due to the addition of an RNAse inhibitor to the internal solution, which raises the osmolarity). Upon conclusion of the recording phase, negative pressure is applied, and the stability of the somatic membrane is monitored using visual and electrophysiological feedback.

The nucleus retrieval stage is focused on attracting the nucleus to the tip of the pipette by using negative pressure and moving the electrode to the location of the nucleus near center of the soma. Patience is key at this point and success relies primarily on visual feedback since the electrode resistance tends to be stable during this stage (*Figure 2H*, time point 1). It is important to maintain constant negative pressure during the transition from the nucleus retrieval to nucleus extraction stage.

The nucleus extraction phase requires slow retraction of the pipette along the same axis of the electrode while maintaining negative pressure. As the nucleus is pulled further from the soma, the cell membrane stretches around the nucleus and ultimately breaks, forming distinct seals around the nucleus (a 'nucleated' patch) and the soma (*Sather et al., 1992*). During this phase, constant monitoring of the membrane seal resistance is critical; the seal formation is observed electrically as a rapid rise in resistance, ideally above 1 GΩ, and referred to as end pipette resistance (endR). It is important that this stage is performed slowly and methodically; achieving the seal can take several minutes (*Figure 2F and H*). *Figure 2G and H* are time series of images and the corresponding test pulse resistance which illustrate the slow pipette retraction with an attached nucleus. The membrane finally breaks and seals between panels 5 and 7, as noted by the sharp rise in resistance shown in *Figure 2H*. After the nucleus has been deposited for RNA sequencing, the fluorescent Alexa dye is viewed conducted to determine if the recorded neuron retained the biocytin fill (last panel in *Figure 2G*).

## Optimizing electrophysiology data quality and throughput

Understanding the intrinsic electrical properties of cortical neurons is a critical component of describing neuronal cell types (*Gouwens et al., 2020*; *Gouwens et al., 2019*; *Scala et al., 2020*; *Ascoli et al., 2008*; *Markram et al., 2015*; *Zeng and Sanes, 2017*; *Tremblay et al., 2016*). A combination of ramp and square step current injection stimulus profiles is effective at revealing discrete electrophysiological neuron types, as well as the continuum of properties between related types (*Gouwens et al.,*

**Figure 1.** Patch-seq is a powerful technique that allows for the characterization of a neuron based on electrophysiology (**E**), transcriptomics (**T**), and morphology (**M**). (**A**) A neuron is patched to characterize the intrinsic properties of a neuron while biocytin and Alexa dye diffuses throughout the soma, dendrites, and axon. At the conclusion of the recording the nucleus is extracted and submitted for RNA-sequencing while the slice is fixed, stained, mounted, and assessed for morphological recovery. (**B**) Exemplar single neuron examples: from left to right are inhibitory neurons from Sst-Cre, Pvalb-Cre, Vip-Cre mice and an excitatory neuron from an Rbp4-Cre mouse. Electrophysiology data shown are the voltage responses to a suprathreshold stimulus to assess a neuron's intrinsic firing properties. Scale bar is 50 mV and 500 ms. The patched neuron was mapped to a common coordinate framework to define anatomical location. Transcriptomic data are shown as a heat map of selected 'on' marker genes detected from the patched neuron. Morphological reconstructions from the patched neurons are shown, as well as their placement within the cortical layer. For inhibitory neurons the darker line represents the dendrite and the lighter line represents the axon. For the excitatory neuron the darker line represents the basal dendrite and lighter line represents the apical dendrite. Data and rates represented here are from the steady-state, fully functional pipeline. Scale bar is 50 mV/500 ms for electrophysiology and 100 µm for reconstructions. Neurons that have a failing morphology outcome but pass for electrophysiology

*Figure 1 continued on next page*

and transcriptomics are salvageable for ET characterization, while neurons that pass all data modalities proceed for MET characterization. Cell IDs are: Sst-Cre – 851016910, Pvalb-Cre – 837594867, Vip-Cre – 755592855, Rbp4-Cre – 841854478.

*2019*). The small differences in properties between types underscores the need for (1) a large dataset to separate related groups and (2) consistent high-quality data. We used the data from this study, now found here https://celltypes.brain-map.org, as a foundation to modify the custom data acquisition system to accommodate online analysis to increase the speed of data acquisition and data quality.

To facilitate flexible patch-clamp data acquisition, we used MIES, a data acquisition package built on top of the Igor Pro software platform. Originally designed to facilitate multichannel synaptic physiology experiments, which require easy multitasking to be performed at scale (*Seeman et al., 2018*), MIES can also be used to acquire intrinsic properties from single neurons (*Gouwens et al., 2019*). We further adapted the MIES software package using custom 'analysis functions' designed to rapidly, automatically acquire valid electrophysiology data sweeps required for cell type classification. We asked two fundamental questions about each sweep: (1) was the baseline prior to stimulus application stable - within and across experiments? and (2) did the cell's membrane potential return to 'baseline' following stimulus application? Only sweeps where both of the above conditions were true were considered for data analysis. Although we can manually exclude data during acquisition when issues arise, we find that most problems can be identified using these two criteria.

To avoid acquiring a sweep with an unstable baseline, we verified that the resting membrane potential (RMP) in a 500 ms pre-stimulus window at the beginning of the experiment was within 1 mV of the target membrane potential. To further determine the stability of the RMP, we used a threshold for the root mean square (RMS) noise of the RMP during the 500 ms baseline evaluation period. If the sweep failed either measure, it was terminated, the bias current was automatically adjusted if necessary, and the sweep was initiated again. If the number of passing sweeps required to pass the stimulus set, plus the number of failed sweeps exceeded the total number of sweeps in the stimulus set, acquisition was terminated, the stimulus set failed and the user was prompted to rerun or abort the experiment. The stimulus set could only proceed as designed once the baseline evaluation passed (*Figure 3A and BFigure 3—figure supplement 3*).

For each sweep, we also ensured that the post-stimulus RMP recovered to baseline (within 1 mV of target voltage) level. The time period for this 'recovery' can vary between cell types and can depend on the intensity of the stimulus so a 'one size fits all' approach, designed to accommodate the longest possible recovery time, is inefficient. To compensate, we designed a recovery period analysis function that would extend to 10 s but shortened once the target RMP was reached. We enforced an absolute 500 ms minimal recovery period immediately after the end of the stimulus, followed by a continuous RMP assay in 500 ms evaluation periods. Once an evaluation period with a recovered baseline was detected, the sweep was completed and considered 'Passed', ultimately completing this sweep in the minimal amount of time (*Figure 3B*, *Figure 3—figure supplement 2*). If none of the evaluation periods passed before the end of the sweep at 10 s, the sweep failed quality control (QC) and was repeated, or the user was prompted. In most instances, a stimulus set ran without triggering the repeat of a sweep (64 % of Ramp, 51 % of Subthreshold Long Square, 70 % of Suprathreshold Long Square, *Figure 3C*). However, a substantial fraction of sweeps in a stimulus set did fail, on average <20 %. If these sweeps had been identified as failing after the conclusion of the experiment, key sweeps may have been lost. By automatically detecting failed sweeps and repeating them sweeps during acquisition, the online analysis functions in MIES maximize the chance of a successful and complete electrophysiology experiment.

Analysis functions can also be designed to change data acquisition during a stimulus period. One such function is the automated detection of action potentials during a ramp stimulus (*Figure 3B*, *Figure 3—figure supplement 3*), followed by the termination of the current injection. This function ends the ramp stimulus after a predefined number (5) of action potentials have been detected, thus increasing efficiency by allowing a more standardized neuronal response. Without the analysis function, the ramp stimulus would continue to depolarize the neuron until it is aborted, ceasing data collection, regardless of when it reached the spiking threshold. With the ramp analysis function, data

**A** Recording
- Sweep level QC
- Accelerated acqusition

**B** Nucleus retrieval
- Centering of pipette
- Slight negative pressure to attract the nucleus

**C** Nucleus extraction
- Slow gradual pipette retraction
- Continued negative pressure
- Observation of seal resistance

**D** cell count vs recording (min)

**E** retrieval (min)

**F** extraction (min)

**G**

**H** test pulse resistance (MΩ) vs time (min)

**Figure 2.** The optimized Patch-seq protocol consists of electrophysiological recording followed by nucleus retrieval and extraction. A schematic of the three major stages of the Patch-seq protocol: (**A**) recording, (**B**) nucleus retrieval, and (**C**) nucleus extraction. Histograms represent the binned time spent for (**D**) recording (N = 7950), (**E**) nucleus retrieval (N = 6750), and (**F**) nucleus extraction (N = 6281). (**G**) Time series of high-resolution images

*Figure 2 continued on next page*

*Figure 2 continued*

showing the gradual pipette retraction with subsequent nucleus extraction. In (**G1**), the red line denotes outline of soma and the last panel is the fill of the soma and processes visualized by Alexa-488. Yellow asterisk identifies the nucleus as it is extracted from the soma. Blue caret identifies the somatic membrane as it is stretched with the extraction of the nucleus. (**H**) is the time plot of steady state resistance, as measured from the test pulse, during the nucleus extraction phase for the neuron in (**G**) with numbers corresponding to brightfield image.

acquisition continues after the stimulus pulse ends, allowing the afterhyperpolarization to be recorded (*Figure 3B*, *Figure 3—figure supplement 3*). Without the analysis function, unabated depolarization may change the state, potentially fatally, of the neuron thereby altering subsequent measurements.

By default, MIES is packaged with the analysis functions described here. The stimuli are linked such that a single click generates the entire dataset in <3 min (*Video 1*, *Figure 3—figure supplement 1*, *Figure 3—figure supplement 2*). The data can be saved directly from MIES as an NWB:N 2.0 (*Ruebel et al., 2019*) file, an emerging, accessible data standard for neurophysiology.

To evaluate how the modified internal solution as well as the rapid data acquisition of the Patch-seq protocol may affect extracted electrophysiological features, we compared data acquired using the Patch-seq protocol (most data from *Gouwens et al., 2020*) with data from *Gouwens et al., 2019*, which was acquired using a more traditional patch clamp approach. On the whole, features extracted from electrophysiology data from three major inhibitory subclass Cre-lines are remarkably consistent. A small number of features show small differences between the datasets, a shift that is consistent with adjustments made to the internal solution composition to compensate for the addition of RNase inhibitor (*Lee et al., 2020*). As described in *Gouwens et al., 2019*; *Gouwens et al., 2020*, a sparse principal components analysis (sPCA) was applied to sets of feature vectors computed from the responses to current steps and pulses. The resulting 45 sPCs were projected onto a two-dimensional space using Uniform Manifold Approximation and Projection (UMAP) (*Becht et al., 2018*). Data from neurons labeled by the three major inhibitory subclass Cre-lines from the combined *Gouwens et al., 2019* (N = 387) and Patch-seq (N = 2220; most data previously reported in *Gouwens et al., 2019*) datasets, shows a high degree of overlap with similar structure (*Figure 3—figure supplement 4B*), along with a subtle shift, consistent with the individual feature analysis.

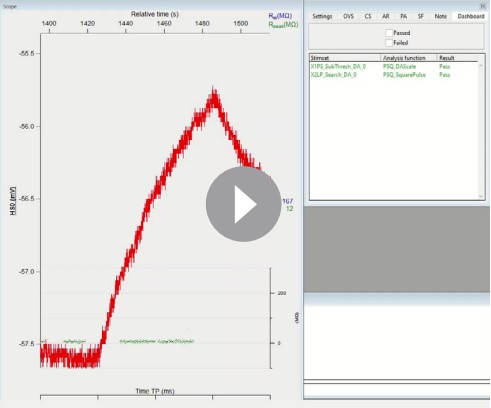

**Video 2.** MIES and the Rheobase analysis function. A video demonstrating the automated functionality of determining rheobase (PSQ_Rheobase) and spike detection. At time point = 0:00 – 0:35, the X3LP_Rheobase_DA_0 stimulus set is ran to search for the Rheobase with a 1 sec square depolarizing pulse. 30, 20, 22, 24, 26, 28, 30 pA steps are administered and ultimately fails to detect an action potential and ultimately determine rheobase - 'Failure as we were note able to find the correct on/off spike pattern'. At time point = 0:52, X3LP_Rheobase_DA_0 ran again with MIES automatically adjusting the current steps to 30, 40, 38, and 40 pA, with successful action potential detection and determination of rheobase.
https://elifesciences.org/articles/65482/figures#video2

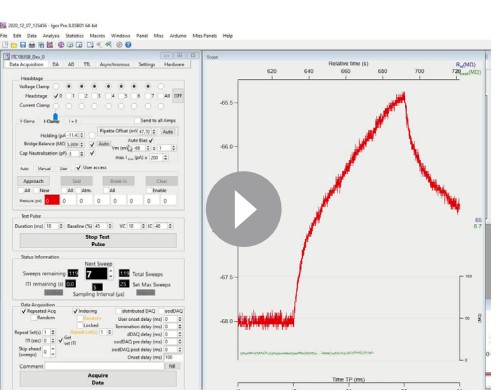

**Video 1.** MIES and a successful experiment. A video demonstrating the efficiency and speed of MIES at acquiring intrinsic features from a whole-cell electrophysiology experiment.
https://elifesciences.org/articles/65482/figures#video1

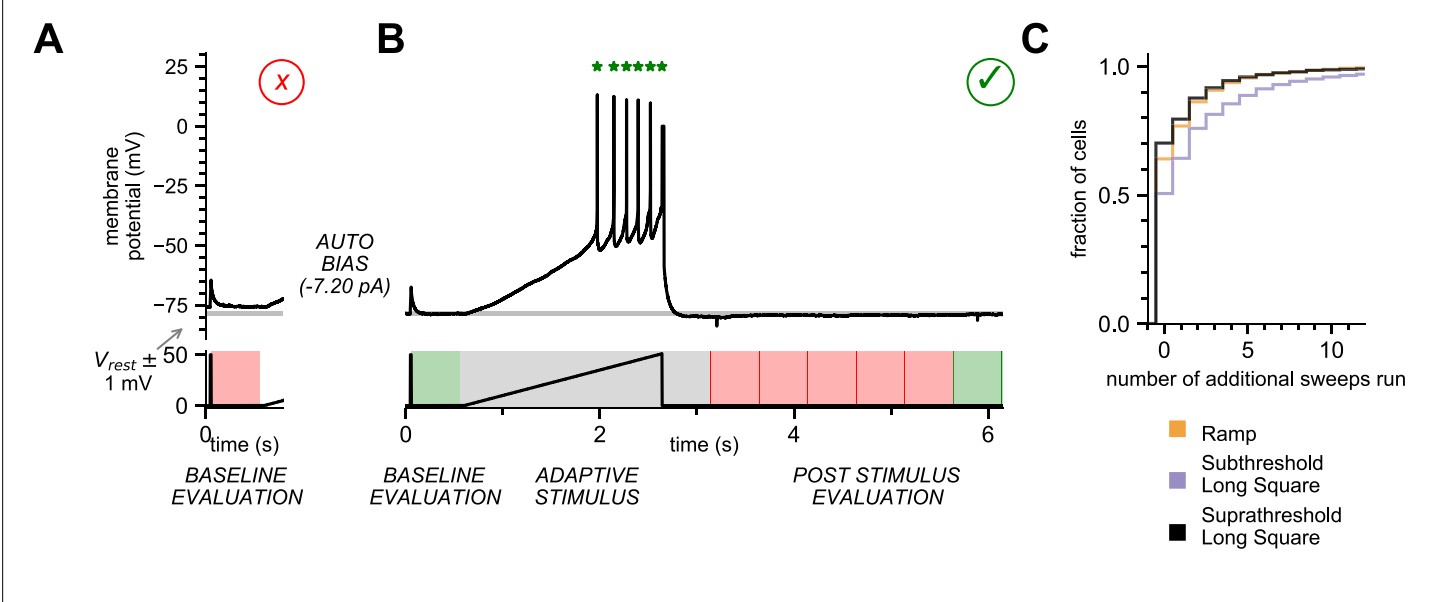

**Figure 3.** Online analysis during electrophysiology recording allows for fast, high-quality data generation. (**A**) Baseline Evaluation failed. Resting Membrane Potential (RMP) was not maintained at ±1 mV (Allen Institute criteria) of the established RMP (gray line at –78 mV); sweep failed and was terminated. Additional autobias current of –7.20 pA was applied during the inter-sweep interval to bring the RMP to the target value of –78 mV. (**B**) Baseline Evaluation passed and the ramp current stimulus was applied (Adaptive Stimulus). Current injection rose at 25 pA/s until five action potentials were detected, at which point current injection was automatically terminated. There was a 500 ms minimal recovery period upon conclusion of stimulus, followed by an assay of RMP every 500 ms (Post Stimulus Evaluation) to test for return to the initial RMP. Once the RMP recovered to within ±1 mV (Allen Institute criteria) acquisition was terminated and sweep was considered as 'passed.' (**C**) Cumulative probability plot showing the number of additional sweeps needed to qualify as passed for each of the three stimulus sets (N = 4488 recordings).

The online version of this article includes the following source data and figure supplement(s) for figure 3:

**Figure supplement 1.** MIES analysis functions for stimulus sets.

**Figure supplement 2.** MIES analysis functions for quality control.

**Figure supplement 3.** MIES dashboard sweep viewer.

**Figure supplement 4.** Consistency of electrophysiological features in Patch-seq and traditional acquisition protocols.

**Figure supplement 4—source data 1.** UMAP coordinates for ME and MET specimens.

**Figure supplement 4—source data 2.** Select electrophysiological features for ME and MET specimens.

## Normalized marker score is a robust measure of transcriptomic data quality

The mRNAs from Patch-seq extractions are often of variable quality, with some samples showing little to no content detected, whereas other samples can match or even exceed the amount of mRNA detected compared to cellular-dissociation based scRNA-seq (*Tripathy et al., 2018*). This variability necessitates the use of a robust, quantitative, and automated measure of mRNA quality that is independent of cell type and that can be used as an indicator for lower quality cells. To address this issue, we utilized a published methodology for assessing transcriptome quality in Patch-seq data sets (*Tripathy et al., 2018*). The premise of this method is that gene expression patterns of cells from matched fluorescence-activated cell sorting (FACS)-based data sets can serve as ground truth profiles for comparison to Patch-seq cells, where Patch-seq cells with technical issues are more likely to diverge from these patterns.

Three metrics are presented for assessing quality, which all rely on defining marker genes for broad cell classes of interest ('on' markers; e.g. Parvalbumin+ interneurons), as well as for cell types that may indicate mRNA contamination from adjacent cell bodies ('off' markers; e.g. astrocytes). First, the normalized marker sum (or NMS) is a ratio of the average expression of 'on' marker genes for a Patch-seq cell relative to the same median expression of these genes in the matched FACS data set for the cell class with highest marker expression. This metric measures the extent to which expected

genes of at least one class are expressed. Second, the contamination score assesses off-target contamination by taking the summed NMS score of all broad cell types (except the assigned class). Higher values of this metric indicate a higher likelihood that mRNA measured from a single neuron also includes RNA from adjacent neurons. Finally, the 'quality score' is a metric aimed at capturing both types of technical issues by measuring the correlation of 'on' and 'off' marker genes in Patch-seq cells with the average expression profile of dissociated cells of the same type.

Here, we expand on the work of *Tripathy et al., 2018* by defining marker genes for class using a more recent study of single neurons collected from mouse primary visual cortex (VISp) and anterolateral motor cortex (ALM) (*Tasic et al., 2018*). We chose 50 marker genes for each class, defining 'on' markers by subclass, and 'off' markers by subclass for non-neuronal cells and by class for neuronal cells. *Figure 4A* displays gene expression data (counts per million and $\log_2$-transformed) and a corresponding high NMS score, from neurons targeted by specific Cre lines and their representative 'on' marker genes. We find that the NMS distribution is roughly bimodal (*Figure 4A*), and we have chosen 0.4 as a rough cut-off for high- and low- quality data (*Gouwens et al., 2020*). Additionally, a low NMS score is more reflective of a lack of detectable genes and not necessarily an increase in 'off' marker expression (genes highlighted in gray in *Figure 4A*). We sought to investigate the relationship of the 'on' marker genes of an assigned subclass and how they relate to neurons that were patched from a matching Cre line. Neurons with a higher NMS score from a Cre line are generally assigned to the appropriate subclass, whereas those with a low NMS score have more promiscuous assignments (*Figure 4B*).

Single-cell transcriptome data from thousands of cells can be used to effectively cluster neurons into transcriptomic types (t-types) (*Tasic et al., 2018*; *Tasic et al., 2016*; *Hodge et al., 2019*). This assignment is classically made using data from dissociated neurons, due to the combination of large cell numbers with high quality transcriptomic data. To probe the morpho-electric properties of these t-types, the transcriptomic data from Patch-seq neurons can be used to map each neuron to a reference dataset (*Gouwens et al., 2020*; *Scala et al., 2020*). To understand how the markers of transcriptome quality predict the ability of the neuron to map to a t-type, we mapped inhibitory neurons from mouse VISp and assessed the quality of mapping described in this manuscript and in *Gouwens et al., 2020* Patch-seq neurons with an NMS <0.4 typically mapped poorly – often to multiple types across multiple sub-types (*Figure 4C*, *Figure 4—figure supplement 1*). However, as described previously (*Gouwens et al., 2020*), a majority of neurons with an NMS ≥ 0.4 mapped to a single type, and those that didn't typically mapped to types within the same subtype (*Figure 4C*, *Figure 4—figure supplement 1*). Some mapping to multiple types is expected, even in FACS data, due to the presence of some neurons identified as 'intermediate' between two types (*Tasic et al., 2018*). As reported previously (*Gouwens et al., 2020*), the reference mapping probability matrix can be used to differentiate between this expected uncertainty due to biological variability, and that which could be due to poor transcriptomic data. Neurons are characterized as highly consistent, moderately consistent, or inconsistent based on the degree of discrepancy from the reference mapping distribution. As expected, neurons with an NMS <0.4 overwhelmingly fall into the 'low consistency' category. In contrast, a majority of neurons with an NMS ≥ 0.4 are mapped in a highly consistent or moderately consistent way. Ultimately, NMS score is very predictive of mapping success, as an NMS of 0.4 allows us to retain the maximum highly/moderately consistently mapping neurons, while accepting relatively few inconsistently mapping neurons (*Figure 4D,E*).

## Extracting the nucleus is key to optimize transcriptomic data quality

Using the marker gene list and NMS score to evaluate transcriptomic quality, we have determined that Patch-seq experiments where we collect the cytosol and nucleus (nucleus+) have significantly higher NMS scores than cytosol-only (nucleus-) samples, t(2803) = 32.2, p < 0.0001 (*Figure 5A*). We designed two sets of metrics to evaluate and confirm the presence of the nucleus in Patch-seq samples: nuclear-specific and subclass-specific gene expression. Gene expression data is a combined measurement of intronic reads (which are localized to the nucleus) and exonic reads (which are found throughout the soma); therefore, nucleus+ samples will have a higher percentage of intronic reads (*Gaidatzis et al., 2015*). Additionally, we examined the presence of Metastasis Associated Lung Adenocarcinoma Transcript 1 (*Malat1*), which has been found as a mammalian-specific nucleus-related gene (*Hutchinson et al., 2007*). Both the intronic reads and *Malat1* gene expression, are correlated with the fraction

**Figure 4.** Normalized marker sum is a measure for transcriptomic quality. To examine the quality of Patch-seq samples we used NMS. The first panel (**A**) is a histogram that displays the range of scores from all neurons from three of the major inhibitory Cre lines (Sst-Cre, Pvalb-Cre, and Vip-Cre) and one excitatory Cre line (Rbp4-Cre) with the dotted line, at 0.4, indicating the separation between pass and fail (N = 2994). The heat maps display the average gene expression of 'on-marker' (in black) or 'off-marker' (in gray) genes for these neurons. Gene expression values are $\log_2$(CPM + 1). (**B**) A heatmap showing how the NMS score relates to the % of cells assigned to a specific subclass and the corresponding Cre line. (**C**) Patch-seq samples above the NMS threshold map with a higher probability to a single reference transcriptomic type (t-type) derived from dissociated (FACS) cells (*Tasic et al., 2018*; mapping procedure described in *Gouwens et al., 2020*). FACS cells also map with high probability to single reference t-types when subjected to the same mapping procedure (N = 646 NMS < 0.4 samples; N = 5175 NMS ≥ 0.4 samples; N = 13464 neuronal FACS samples). (**D**) Sankey plot showing the confidence with which Patch-seq transcriptomes with NMS above or below 0.4 mapped to one or more reference t-types derived from dissociated cells (mapping confidence measure described in *Gouwens et al., 2020*). (**E**) ROC analysis comparing N = 4627 highly and moderately consistently mapping versus N = 1194 inconsistently mapping patched neurons. Heat map of the ROC curve is the NMS. Gray circle is at NMS = 0.4 and gray dashed line is the line of identity.

The online version of this article includes the following source data and figure supplement(s) for figure 4:

**Source data 1.** Select genes and corresponding values for cell specimens obtained from Pvalb-IRES-Cre;Ai14 mice.

**Source data 2.** Select genes and corresponding values for cell specimens obtained from Rbp4_KL100-Cre;Ai14 mice.

**Source data 3.** Select genes and corresponding values for cell specimens obtained from Sst-IRES-Cre;Ai14 mice.

**Source data 4.** Select genes and corresponding values for cell specimens obtained from Vip-IRES-Cre;Ai14 mice.

**Source data 5.** Percentage values of cells assigned to a specific subclass and NMS score for Sst-Cre, Pvalb-Cre and Vip-Cre mice.

**Source data 6.** Probability of single cell's ability to map to a single cluster.

**Figure supplement 1.** Mapping probabilities for neurons with low/high normalized marker sum.

*Figure 4 continued on next page*

*Figure 4 continued*

**Figure supplement 1—source data 1.** Probability of mapping to transcriptomic cell types for FACS specimens.

**Figure supplement 1—source data 2.** Probability of mapping to transcriptomic cell types for NMS-fail Patch-seq specimens.

**Figure supplement 1—source data 3.** Probability of mapping to transcriptomic cell types for NMS-pass Patch-seq specimens.

of RNA collected from the nucleus relative to the cytoplasm (*Bakken et al., 2018*). Indeed, nucleus+ Patch seq samples had a higher correlation with *Malat1* expression and the percentage of reads mapped to introns compared to nucleus- samples (*Figure 5B*). These measures demonstrate that nucleus+ samples have higher transcriptomic quality and can also serve as confirmatory evidence for collection of nuclei.

In addition to intronic reads and nucleus-specific genes, nucleus+ samples have a higher detection of genes overall relative to nucleus- samples (*Figure 5—figure supplement 1A*). More specifically the nucleus+ samples have a higher fraction of 'on' marker gene expression specific to the appropriate Cre lines. In contrast 'off' marker genes, genes not specific to the Cre line, or glia-related genes, are less differentiated between nucleus+ and nucleus- samples (*Figure 5C*). The histograms to the right of each scatter plot quantify the average subclass-specific gene expression for nucleus+ samples and how they relate to their FACS (gray) counterparts. One important item to note is the higher detection of glia-related genes in Patch-seq samples compared to FACS. This is expected due to the inherent nature of the Patch-seq process as the patch pipette navigates through the tissue. We can conclude from these plots that the detection of subclass-specific gene expression in nucleus+ samples is similar to that of FACS-isolated samples.

Due to the sensitivity and potential instability of the neuron in the dialyzed whole-cell configuration, we sought to determine if there was a relationship between the experiment duration and the quality of extracted mRNA and subsequent cDNA (*Sucher et al., 2000*; *Veys et al., 2012*). To do this, we tracked the time spent at each of the Patch-seq stages (*Figure 5D and E*). We found that time had no effect on the quality of the transcriptomic content, as assessed with NMS. As shown in *Figure 5A and D*, nucleus+ conditions have a significantly higher NMS with no effect of patch duration. Additionally, the time of nucleus retrieval and extraction phases did not affect NMS. This corroborates other findings (*Cadwell et al., 2017*) that the integrity of the RNA is preserved during the Patch-seq recordings. Note that that the extraction times can range from less than 1 minute up to 8 minutes to successfully extract the nucleus. Despite the lack of an effect, it may still be advisable to keep the duration shorter for other reasons such as increasing throughput or optimizing morphological recovery. In addition to the patch duration, we also investigated the role of the initial whole-cell patch clamp parameters (access resistance to the neuron and depth of neuron within the slice) to determine a role in the quality of the Patch-seq sample. Surprisingly, there was no effect with the initial access resistance or neuron depth on NMS (*Figure 5—figure supplement 1*).

Additionally, the cDNA can be evaluated to inform the quality of the Patch-seq samples. Electrophoretograms are obtained from a fragment or bioanalyzer and can provide unique metrics about the amplified cDNA. Quantitatively, two parameters can be obtained: (1) the amount of amplifiable content, and (2) the quality of the cDNA, measured as the ratio of high base pair to total material. Qualitative analysis can provide confirmation about the quantitative data obtained, such as proper distributions and shape of the electrophoretogram, positive controls, and artifacts. We found that NMS score is highly correlated with each metric and that there is a clear separation between nucleus+ and nucleus- samples. The difference between the two outcomes (nucleus+ vs. nucleus-) was found to be highly significant for each metric: cDNA quantity, t = 11.53, p < 0.0001 and cDNA quality, t = 18.84, p < 0.0001 (*Figure 5—figure supplement 2A*). Receiver operating characteristic (ROC) curve analyses were performed to evaluate the sensitivity and specificity of the cDNA for the presence or absence of the nucleus in Patch-seq samples. As shown in Figure 5—figure supplement 2, both the quantity and quality of the cDNA achieved an area under the curve (AUC) was ≥0.80, demonstrating that the presence of the nucleus is strongly associate with high-quality data.

## Optimizing morphology success with Patch-seq recordings

The shape and morphological features of a neuron are important for its function and can be used to classify and define types (*Zeng and Sanes, 2017*; *Gouwens et al., 2019*; *Harris and Shepherd,*

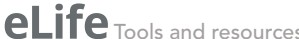

**Figure 5.** Nucleus extraction is key to transcriptomic success. (**A**,**B**,**D**) examine an early dataset of Patch-seq samples with both nucleus+ and nucleus- outcomes (N = 1908 nucleus+ samples; N = 897 nucleus- samples). The bar plots in (**A**) represent the difference in NMS scores. (**B**) A scatter plot viewing the distribution of % reads mapped to introns and how they related to the NMS score for nucleus+ (top) and nucleus- (bottom). Color ramp indicates the % reads mapped to Malat1. (**C**) Scatter plots examining the relationship between nucleus+ or nucleus- and the presence of subclass-specific marker gene expression in a subset of data from (A,B,D) that includes four different Cre lines. The colors of dots represent the subclass to which the gene is specific (N = 50 Sst, N = 50 Pvalb, N = 50 Vip, N = 193 Excitatory, N = 146 Glia genes). Data pooled from N = 386/131 Sst-Cre, N = 97/55 Pvalb-Cre, N = 204/95 Vip-Cre and N = 84/38 Rbp4-Cre neurons (nucleus+/nucleus-). The histograms are the quantification of subclass-specific genes for each of the respective Cre lines for nucleus+ samples only. The gray bars are FACS data (N = 356 Sst-Cre, N = 532 Pvalb-Cre, N = 345 Vip-Cre and N = 771 Rbp4-Cre). In (**D**), the histograms represent the binned time spent and the neuron count for recording duration, the darker bars plot successful extraction of the nucleus (nucleus+) whereas the lighter bars plot failed extraction of nucleus (nucleus-). Solid line represents the NMS score for nucleus+ samples as a function of time, whereas gray dotted line represents nucleus- samples. (**E**) represents the binned time spent and neuron count for nucleus retrieval (left) and nucleus extraction (right) (N = 5930). The solid line represents the NMS score as a function of time.

The online version of this article includes the following source data and figure supplement(s) for figure 5:

**Source data 1.** Subclass-specific gene expression for nucleus+ and nucleus- specimens from Sst-Cre, Pvalb-Cre, Vip-Cre and Rbp4_KL100_Cre mice.

**Figure supplement 1.** Initial access resistance and cell depth in slice are not associated with different transcriptomic or morphological outcomes.

**Figure supplement 2.** Nucleus+ is predictive for successful sequencing and cDNA data.

---

*2015*; *Markram et al., 2015*; *Lodato and Arlotta, 2015*). Obtaining morphologies from Patch-seq samples has been a challenge and strategies for improving recovery have not been thoroughly examined. Early Patch-seq studies were unable to recover the morphology of the patched neuron and had to rely on electrophysiological classifiers to infer the morphological properties (*Fuzik et al., 2015*;

---

*Cadwell et al., 2015*). More recently, *Cadwell et al., 2017* have described success in recovering morphologies from Patch-seq samples using longer recording times but RNA quality of these neurons was generally lower than reported for FACS-isolated neurons. We have shown that extracting the nucleus (nucleated-patch) is key for transcriptomic success; historically, this paradigm has been used for studies of membrane biophysics. In previous studies using nucleated patches for this purpose (*Eyal et al., 2016*; *Bekkers, 2000*; *Gurkiewicz and Korngreen, 2006*), morphological recovery of the neuron has not been well studied. Here we created standard metrics to measure the neuron morphology outcomes, then used those evaluations to adjust the neuron recording protocol to optimize morphological recovery while retaining high-quality RNA extraction.

*Figure 6A–E* displays representative examples of the biocytin fill quality calls with the Patch-seq technique. 'High quality' fills have a visible soma and processes and are suitable for 3D digital reconstructions as shown in *Figure 6A and B* for an excitatory and inhibitory neuron, respectively. 'Insufficient axon' fills represent neurons with filled dendrites, but weakly filled axons or axons that are orthogonal and exit the slice (*Figure 6C*). 'Medium quality' fills have a visible soma and some processes but are not suitable for 3D digital reconstruction (*Figure 6D*). 'Failed fills' have no visible somas and likely result from the collapse of the soma during nucleus extraction and subsequent leakage of the biocytin (*Figure 6E*). A single coronal slice with multiple Patch-seq recordings can lead to varying outcomes for morphological quality (*Figure 6—figure supplement 1*), suggesting that factors prior to the slice processing phase are critical determinants of morphological recovery. Here, we focused on the impact of recording variables on morphological recovery outcomes.

Many studies have shown that patch duration must range from 15 mins up to 60 mins to obtain optimal filling of neuronal processes with biocytin (*Gouwens et al., 2019*; *Marx et al., 2012*; *Cadwell et al., 2017*). To maximize throughput, we targeted a recording duration <10 mins. Surprisingly, we found no effect of the recording duration (range: 3–12 min) on the fraction of neurons that were deemed optimal for morphology reconstruction (*Figure 6F*). The duration dependence of morphology output was mostly flat for other phases of the recording process (*Figure 6G and H*), with perhaps a slight upward trend in outcome for longer retrieval times (*Figure 6G*). To investigate the effect that recording multiple neurons per slice has on morphology outcomes, we compared the morphologies of neurons from brain slices that had 1, 2, 3, or 4 recordings before slice fixation. We found that multiple Patch-seq recordings could be obtained in a single slice with no deleterious effects as the prior patched neuron(s) remained intact during subsequent recordings. We did observe a trend in which the last neuron recorded in the slice had the poorest outcome for 'high quality' with an increase in 'insufficient axon' or 'failed' outcomes (*Figure 6—figure supplement 3A*), indicating the possibility of insufficient time for complete diffusion of biocytin throughout the neuron prior to fixation. This is consistent with a previous report (*Cadwell et al., 2017*) showing that when the slice is fixed immediately after a single neuron is recorded, longer recording durations are required for sufficient fill.

We next asked how the resistance of the nucleated patch membrane (the end pipette resistance, endR, *Figure 2H*) predicts the ultimate neuron morphology, as it likely reflects how well the membrane reseals around both the extracted nucleus and the neuron remaining in the slice. We found the endR to be highly predictive of the final morphology outcome for both excitatory and inhibitory neurons. When comparing the outcome of high and medium quality fills (combined) versus failed fills using a ROC curve, we find an area under the curve (AUC) of 0.56 and 0.62 for excitatory and inhibitory neurons, respectively. Most interestingly, there is a prominent shoulder in the ROC curve at an endR of 100 MΩ. Using this value as a cutoff can reduce the fraction of morphology failures by about 30 % at a cost of <5 % of high and medium morphologies for both excitatory and inhibitory neurons (*Figure 6I and J*). EndR did not impact the insufficient filling of the neuron, which is likely influenced by the recording duration or positioning of the neuron within the slice (*Figure 6K*). This demonstrates that recordings that end with a pipette with a high endR are more successful at retaining the morphological fill. Access resistance during a recording may reflect the continuity of the internal solution between the electrode and cell. Interestingly, we found that the value of that electrical access does not relate to the cell's morphology outcome (*Figure 5—figure supplement 1*).

Unsurprisingly, cell health is also a significant factor in the ability to retain the morphology of the recorded neuron. We performed a qualitative rating on a scale of 1 (worst) to 5 (best) of cell health. This rating is multifaceted including a visual assessment, using metrics such as soma shape and sharpness of the plasma membrane (*Figure 6—figure supplement 2*), and a recording quality assessment,

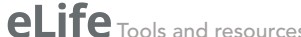

**Figure 6.** End pipette resistance, but not recording duration, is a significant predictor of morphology success. Example biocytin recovery and subsequent morphological reconstructions of a high quality filled Rbp4-Cre+ excitatory (**A**) and a Vip-Cre+ inhibitory (**B**) neuron. (**C**) A Vip-Cre+ inhibitory neuron that failed due to an insufficiently filled axon. (**D**) An interneuron that was weakly filled and was classified as medium quality. (**E**) A failed fill where no processes are visible and biocytin leakage is apparent. (**A–E**) are high-resolution 63 x stack minimum image projection (MIP) images

*Figure 6 continued on next page*

*Figure 6 continued*

and (**A–C**) have the corresponding morphological reconstruction with dendrites in red and axon in blue. Line plots demonstrate the time spent for each of the Patch-seq phases: (**F**) recording (N = 1666 excitatory; N = 4214 inhibitory), (**G**) nucleus retrieval (N = 1514 excitatory; N = 4040 inhibitory), and (**H**) nucleus extraction (N = 1537 excitatory; N = 3752 inhibitory), and how they relate to the morphological recovery. ROC analyses comparing high/medium quality versus failed morphology outcomes for patched (**I**) excitatory (N = 1064 high/medium quality; N = 531 failed) and (**J**) inhibitory neurons (N = 1121 high quality; N = 1288 failed). (**K**) ROC analysis comparing N = 1121 high quality versus N = 1041 insufficient axon morphology outcomes for patched inhibitory neurons. Heat map of the ROC curve is the end pipette resistance (MΩ) measured at the conclusion of nucleus extraction. Gray dashed line in ROC analyses is the line of identity. Cell IDs are: high-quality excitatory neuron – 841854478, high-quality inhibitory neuron – 755592855, insufficient axon – 642494910.

The online version of this article includes the following figure supplement(s) for figure 6:

**Figure supplement 1.** Example biocytin recovery and morphological calls.

**Figure supplement 2.** Examples of cell health ranking.

**Figure supplement 3.** Patching sequence and cell health and how it relates to morphological recovery.

**Figure supplement 4.** Examples of cell health and morphological outcome examples.

using metrics such as baseline stability and number of failed sweeps. We found that cells ranked one or 2, had a lower chance of a 'high quality' score than cells with rank of 3, 4, or 5. Interestingly, there was little to no improvement of 'high quality' fills between rankings of 3 and 5 (*Figure 6—figure supplement 3B*). *Figure 6—figure supplement 4* shows representative 63 x resolution minimum image projections (MIPs) of biocytin fills and their resultant morphological calls compared to the cell health assessment score.

## Application to other species and cell types

A powerful aspect of the FACS-based transcriptomic studies is the ability to use similar methods to profile and compare neurons from different brain regions or species (*Bakken et al., 2018*; *Hodge et al., 2019*; *Kalmbach et al., 2020*). With that in mind, we co-developed this protocol in both mouse tissue and tissue from human surgical resections. We sampled 103 human neocortical neurons across the four lobes (*Figure 7—figure supplement 1A*), 330 of which were described in a previous study (*Berg et al., 2020*; *Figure 1—figure supplement 1*). The expanded dataset used for the analysis presented here includes neurons that did not pass the stringent quality control (QC) metrics required to be part of that previous study in addition to human inhibitory neurons to demonstrate the generalizability of the metrics and protocol to different neocortical cell types. We observed a pass rate of 76%, 77%, and 47% for electrophysiology, transcriptomics and morphology, respectively; with an ultimate final rate of 69 % for ET and 28 % for MET characterization (*Figure 7—figure supplement 1B*) – all of which are comparable to mouse rates of success. Importantly, the key predictors of success – collecting the nucleus for transcriptomics and end electrode resistance for morphology, were consistent between mouse and human experiments (*Figure 7—figure supplement 1C and D*). In one example, we recorded from 10 neurons in the same human cortical slice (*Figure 7*) – for each neuron, the nucleus was extracted, and the end electrode resistance was >1 GΩ. The fact that a single human slice can yield 100 % success despite lower population averages highlights that higher success rates are feasible, especially in human tissue, which has several experimental variables such as age of the patient, preexisting conditions, etc. Gene expression was consistent with the mapped transcriptomic types (*Figure 7C*) and morphoelectric properties of each neuron are consistent with the cell types to which they are mapped. Glutamatergic neurons are pyramidal and have regular, adapting spiking patterns, while the GABAergic neurons show a greater diversity in their morphologies and firing patterns. To further demonstrate the generalizability of the approach, we targeted pyramidal neurons in macaque ex vivo acute brain slices from the temporal cortex region. We collected high-quality, morphoelectric, and transcriptomic data from 36 neurons that show that quality metrics were similarly correlated with nucleus extraction (*Figure 7—figure supplement 2*) as shown with mouse and human Patch-seq experiments.

## Discussion

We have optimized the Patch-seq technique using a standardized approach to provide a comprehensive protocol and guidance for others in the community. These findings build upon key components of

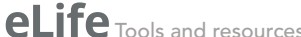

**Figure 7.** Exemplar human neocortical slice with 10 successful Patch-seq recordings. (**A**) is a 20 x bright field image of a human acute neocortical slice containing biocytin reacted fills with overlaid digital reconstructions from ten Patch-seq recordings. Reconstructions are colored by their corresponding mapped subclass: IT (L2-3) - blue, IT (L4-6) - green, SST - orange, VIP - purple and PVALB - pink. Axons are light-weighted lines and the basal/apical dendrites for excitatory neurons are distinguishable by darker and lighter colors, respectively. Neurons are numbered in the order in which they were

*Figure 7 continued on next page*

patched. Scale bar is 500 µm. (**B**) Voltage responses of corresponding neuron to 1 s-long current steps equal to –70 pA and rheobase (bottom traces) and rheobase +80 pA (top trace). Scale bar, vertical 80 mV, horizontal 500 ms. (**C**) Dot plot of marker genes that are differentially expressed across human cortical neurons. Expression values (color) for each neuron in slice pictured in (**A**) are log$_2$-transformed. Percentages calculated from N = 3706 IT (L2-3); N = 5636 IT (L4-6); N = 809 SST; N = 1298 VIP; N = 809 PVALB dissociated neurons. Cell IDs, from left to right are: 689300528, 689306781, 689309060, 689371684, 689312221, 689319155, 689323433, 689331391, 689338841, 689360084.

**Figure supplement 1.** Application of the Patch-seq protocol to human neocortical cells.

**Figure supplement 2.** Application of the Patch-seq protocol to macaque neocortical neurons.

previous detailed Patch-seq protocols (*Cadwell et al., 2017*; *Fuzik et al., 2015*) to increase throughput using custom software by identifying key predictors of data quality, most critically extracting the nucleus and maintaining membrane integrity as the electrode is retracted. The large size of the dataset (combined mouse and human N = 9664) has allowed us to examine the relationship between specific QC metrics and experimental outcomes. We have provided a guide for each metric including the range and what type of data to expect. For example, due to the high costs of sequencing, it might be in researchers' financial interests to use the cDNA metrics and be judicious about which samples to send for sequencing. More importantly, these metrics can and allow users to make predictions and to optimize approaches for specific applications.

This protocol has resulted in the successful collection of large-scale Patch-seq data from mouse and human brain neuron types, and these data are publicly accessible as part of the Allen Cell Types Database (https://celltypes.brain-map.org/) as well as the NIH's Brain Research through Advancing Innovative Neurotechnologies (BRAIN) Initiative - Cell Census Network (BICCN, https://biccn.org). To accelerate data collection in the scientific community, we identified two use cases for an independent researcher who may want to generate comparable data:

1. Determining the morphoelectric phenotype of neurons corresponding to a large (FACS-based) transcriptomics dataset. The Patch-seq platform makes morphoelectric data accessible to a transcriptomics community that is accustomed to using large datasets to probe biological variance within a population. However, previous descriptions of the Patch-seq method have detailed a lower-throughput technique, suitable for targeted small studies (*Lipovsek et al., 2020*; *Cadwell et al., 2017*). Here, we describe an approach that can be used to generate combined morphoelectric and transcriptomic data at a scale that approaches recent transcriptomic studies. Indeed, the number of Patch-seq neurons in a recent manuscript using this protocol (4270 neurons, *Gouwens et al., 2020*) was more than 2.5 times the number of neurons in a FACS-based single-cell transcriptomics study just a few years ago (1679 cells, *Tasic et al., 2016*).
2. A smaller study that 'extends' the Allen Institute data sets to answer a specific research question. Building upon large, standardized datasets has been a successful way to address challenging questions (*Campbell et al., 2017*; *Steinmetz et al., 2019*). However, in the case of Patch-seq, combined analysis of electrophysiological data generated in different labs has proved challenging to integrate (*Tripathy et al., 2018*). To combat this, here we detail both the approach and any quality metrics we use to exclude data. Another advantage using the tools described here is that MIES saves data in NWB:N 2.0, the open data format adopted by a number of labs, including all electrophysiology data deposited in the Allen Cell Types Database and BICCN. Pre-saved stimulus sets and the ability to save data in a standardized format will facilitate combined analysis of new and archived data.

Despite the improvements described here, there is room for additional optimization of the Patch-seq technique, primarily in morphological recovery, where ~ 46 % of experiments result in neurons fit for digital reconstruction. Since morphological recovery is related to the qualitative ranking of cell health, additional optimization of the tissue slicing protocol tailored to specific mouse age, region, or cell types (*Ting et al., 2018*) could lead to improved recovery. An additional area for further investigation is the effect of the size of the electrode on electrophysiological, transcriptomic, and morphological data quality. Since the seal resistance following retraction is a strong predictor of morphology outcome, it may be that smaller electrodes, which would disturb less of the total cell membrane, would be predicted to improve morphological recovery success. However, we initially found that smaller electrodes made it difficult to collect the nucleus, so we did not pursue this strategy

further. Depending on the scientific question being asked, one can make different trade-offs between potentially conflicting data modalities.

To improve throughput, the patch-clamp process could be further automated (*Kodandaramaiah et al., 2016*, *Holst et al., 2019*). Indeed, here we have shown that automated analysis of electro-physiological features improves both the speed of acquisition and the quality of the ultimate data set. Automation could improve throughput by allowing the user to focus on getting the next neuron while the automated patch-clamp device initiates a recording. Implementing automation could also improve the data quality in other modalities in Patch-seq. For example, automated retraction of the pipette during nucleus extraction, with endR feedback, may result in a slower, more standardized movement with the potential to improve the rate of morphological recovery.

The protocol described here is focused on the interrogation of each neuron's intrinsic electrical properties, but an obvious extension would be to incorporate Patch-seq with measurement of synaptic properties. Characterizing the connectivity and synaptic dynamics has been shown to differentiate classes of neurons (*Jiang et al., 2015*; *Seeman et al., 2018*; *Földy et al., 2016*) and linking the rates or strength of synaptic connections with genes of interest and subsequent transcriptomic types would further our understanding of the functional role of cell types.

Large-scale transcriptomic studies have led to an incredible amount of progress in understanding the degrees of cell type differences between brain regions (*Tasic et al., 2018*; *Yao et al., 2021*) and species (*Bakken et al., 2018*; *Hodge et al., 2019*; *Bakken et al., 2016*). To probe the phenotypic consequences of this differentiation, a platform is required that is robust to these dimensions. We have successfully used the Patch-seq protocol described here to study non-human primate and human excitatory neurons (*Berg et al., 2020*; *Kalmbach et al., 2020*; *Bakken, 2020*), mouse cortical neurons (*Gouwens et al., 2020*; *Keaveney et al., 2020*), and hippocampal (*Dudok et al., 2021*) interneurons. Here, we demonstrate that the key to successfully adapting the protocol to a new system, like the human or non-human primate, is to focus on consistency in a few aspects, including nucleus extraction and maintaining membrane integrity during retraction. By dividing the protocol into segments with clear metrics and benchmarks, other researchers can adapt critical parts of the protocol while monitoring trade-offs in other modalities. Intriguingly, in some human cases, we can achieve high-quality triple modality data from many neurons in a single slice.

As FACS-based single-cell transcriptomics establishes a foundation for studying neuronal cell types using large, standardized datasets with common tools, complementary techniques must adapt to facilitate faster data acquisition that can be applied to diverse tissues, along with transparent quality metrics. Patch-seq, although a relatively young technique, has shown scalability that holds promise for accelerating progress toward understanding of the phenotypic consequences of transcriptomic variability. The protocol described here details the keys to that scalability and provides the tools that allow others to adapt the approach for their own diverse research programs.

# Materials and methods

**Key resources table**

| Reagent type (species) or resource | Designation | Source or reference | Identifiers | Additional information |
|---|---|---|---|---|
| Strain, strain background (*M. musculus*) | B6.Cg-Gt(ROSA)$^{26Sortm14(CAG-tdTomato)Hze}$/J, Ai14 | The Jackson Laboratory | RRID: IMSR_JAX: 007914 | |
| Strain, strain background (*M. musculus*) | B6; Batf3$^{tm1(flpo)Hze}$/J, Batf3-IRES2-FlpO-neo | The Jackson Laboratory | RRID: IMSR_JAX: 034301 | |
| Strain, strain background (*M. musculus*) | B6.Cg-Calb1$^{tm2.1(cre)Hze}$/J, Calb1-IRES2-Cre | The Jackson Laboratory | RRID: IMSR_JAX: 028532 | |
| Strain, strain background (*M. musculus*) | STOCK Cck$^{tm1.1(cre)Zjh}$/J, Cck-IRES-Cre | The Jackson Laboratory | RRID: IMSR_JAX:012706 | |

*Continued on next page*

*Continued*

| Reagent type (species) or resource | Designation | Source or reference | Identifiers | Additional information |
|---|---|---|---|---|
| Strain, strain background (*M. musculus*) | B6;129S6-Chat^tm2(cre)Lowl^/J, Chat-IRES-Cre | The Jackson Laboratory | RRID: IMSR_JAX:006410 | |
| Strain, strain background (*M. musculus*) | STOCK Tg(Chrna2-cre) OE25Gsat/Mmucd, Chrna2-Cre_OE25 | MMRRC | RRID: MMRRC_036502-UCD | |
| Strain, strain background (*M. musculus*) | B6; Chrna6^tm1(flpo)Hze^ /J, Chrna6-IRES2-FlpO-neo | The Jackson Laboratory | RRID: IMSR_JAX:034302 | |
| Strain, strain background (*M. musculus*) | B6.Cg-Ccn2tm1.1(folA/cre)Hze/J, Ctgf-T2A-dgCre | The Jackson Laboratory | RRID: IMSR_JAX:028535 | |
| Strain, strain background (*M. musculus*) | B6;129S6-Ctxn3^em1(flpo)Hze^/J, Ctxn3-IRES2-FlpO-neo | The Jackson Laboratory | RRID: IMSR_JAX:034303 | |
| Strain, strain background (*M. musculus*) | B6(Cg)-Cux2^tm3.1(cre/ERT2)Mull^/Mmmh, Cux2-CreERT2 | MMRRC | RRID: MMRRC_032779-MU | |
| Strain, strain background (*M. musculus*) | B6;129S-Esr2^tm1.1(cre)Hze^/J, Esr2-IRES2-Cre | The Jackson Laboratory | RRID: IMSR_JAX: 030158 | |
| Strain, strain background (*M. musculus*) | B6(Cg)-Etv1^tm1.1(cre/ERT2)Zjh^/J, Etv1-CreERT2 | The Jackson Laboratory | RRID: IMSR_JAX:013048 | |
| Strain, strain background (*M. musculus*) | B6J.Cg-Gad2^tm2(cre)Zjh^/MwarJ, Gad2-IRES-Cre | The Jackson Laboratory | RRID: IMSR_JAX:028867 | |
| Strain, strain background (*M. musculus*) | STOCK Tg(Gad1-EGFP)94Agmo/J, Gad67-GFP_X94 | The Jackson Laboratory | RRID: IMSR_JAX:006334 | |
| Strain, strain background (*M. musculus*) | STOCK Tg(Colgalt2-cre) NF107Gsat/Mmucd, Glt25d2-Cre_NF107 | MMRRC | RRID: MMRRC_036504-UCD | |
| Strain, strain background (*M. musculus*) | B6;129S6-Gpr139^tm1.1(flpo)Hze^/J, Gpr139-IRES2-FlpO-WPRE-neo | The Jackson Laboratory | RRID: IMSR_JAX:034304 | |
| Strain, strain background (*M. musculus*) | STOCK Tg(Htr3a-cre)NO152 Gsat/Mmucd, Htr3a-Cre_NO152 | MMRRC | RRID: MMRRC_036680-UCD | |
| Strain, strain background (*M. musculus*) | B6.Cg-Ndnf^tm1.1(folA/cre)Hze^/J, Ndnf-IRES2-dgCre | The Jackson Laboratory | RRID: IMSR_JAX:028536 | |
| Strain, strain background (*M. musculus*) | STOCK Nkx2-1^tm1.1(cre/ERT2)Zjh^/J, Nkx2.1-CreERT2 | The Jackson Laboratory | RRID: IMSR_JAX:014552 | |
| Strain, strain background (*M. musculus*) | B6;129S-Nos1^tm1.1(cre/ERT2)Zjh^/J, Nos1-CreERT2 | The Jackson Laboratory | RRID: IMSR_JAX:014541 | |
| Strain, strain background (*M. musculus*) | B6;129S-*Npr3*^tm1.1(cre)Hze^/J, Npr3-IRES2-Cre | The Jackson Laboratory | RRID: IMSR_JAX: 031333 | |

*Continued on next page*

*Continued*

| Reagent type (species) or resource | Designation | Source or reference | Identifiers | Additional information |
|---|---|---|---|---|
| Strain, strain background (*M. musculus*) | FVB-Tg(Nr5a1-cre)2Lowl/J, Nr5a1-Cre | The Jackson Laboratory | RRID: IMSR_JAX:006364 | |
| Strain, strain background (*M. musculus*) | B6.FVB(Cg)-Tg(Ntsr1-cre) GN220Gsat/Mmucd, Ntsr1-Cre_GN220 | The Jackson Laboratory | RRID: MMRRC_030648-UCD | |
| Strain, strain background (*M. musculus*) | B6;129S-*Oxtr*<sup>tm1.1(cre)Hze</sup>/J, Oxtr-T2A-Cre | The Jackson Laboratory | RRID: IMSR_JAX: 031303 | |
| Strain, strain background (*M. musculus*) | B6;129S-*Pdyn*<sup>tm1.1(cre/ERT2)Hze</sup>/J, Pdyn-T2A-CreERT2 | The Jackson Laboratory | RRID: IMSR_JAX: 030197 | |
| Strain, strain background (*M. musculus*) | B6;129P2-Pvalb<sup>tm1(cre)Arbr</sup>/J, Pvalb-IRES-Cre | The Jackson Laboratory | RRID: IMSR_JAX:008069 | |
| Strain, strain background (*M. musculus*) | B6.Cg-Pvalb<sup>tm1.1(cre/ERT2)Hze</sup>/J, Pvalb-T2A-CreERT2 | The Jackson Laboratory | RRID: IMSR_JAX: 021189 | |
| Strain, strain background (*M. musculus*) | STOCK Tg(Rbp4-cre)KL100Gsat/ Mmucd, Rbp4-Cre_KL100 | MMRRC | RRID: MMRRC_031125-UCD | |
| Strain, strain background (*M. musculus*) | B6.Cg-Rorb<sup>tm3.1(flpo)Hze</sup>/J, Rorb-IRES2-Cre | The Jackson Laboratory | RRID: IMSR_JAX: 029590 | |
| Strain, strain background (*M. musculus*) | B6;C3-Tg(Scnn1a-cre)1Aibs/J, Scnn1a-Tg1-Cre | The Jackson Laboratory | RRID: IMSR_JAX: 009111 | |
| Strain, strain background (*M. musculus*) | B6;C3-Tg(Scnn1a-cre)2Aibs/J, Scnn1a-Tg2-Cre | The Jackson Laboratory | RRID: IMSR_JAX:009112 | |
| Strain, strain background (*M. musculus*) | B6;C3-Tg(Scnn1a-cre)3Aibs/J, Scnn1a-Tg3-Cre | The Jackson Laboratory | RRID: IMSR_JAX:009613 | |
| Strain, strain background (M. musculus) | STOCK Tg(Sim1-cre)KJ18Gsat/ Mmucd, Sim1-Cre_KJ18 | MMRRC | RRID: MMRRC_031742-UCD | |
| Strain, strain background (M. musculus) | B6J.129S6(FVB)-Slc17a6<sup>tm2(cre)Lowl</sup>/MwarJ, Slc17a6-IRES-Cre | The Jackson Laboratory | RRID: IMSR_JAX:028863 | |
| Strain, strain background (M. musculus) | B6;129S-Slc17a7<sup>tm1.1(cre)Hze</sup>/J, Slc17a7-IRES2-Cre | The Jackson Laboratory | RRID: IMSR_JAX:023527 | |
| Strain, strain background (M. musculus) | STOCK Tg(Slc17a8-icre)1Edw/ SealJ, Slc17a8-iCre | The Jackson Laboratory | RRID: IMSR_JAX:018147 | |
| Strain, strain background (M. musculus) | B6;129S-*Slc17a8*<sup>tm1.1(cre)Hze</sup>/J, Slc17a8-IRES2-Cre | The Jackson Laboratory | RRID: IMSR_JAX: 028534 | |
| Strain, strain background (M. musculus) | B6.Cg-*Slc32a1*<sup>tm1.1(flpo)Hze</sup>/J, Slc32a1-IRES2-FlpO | The Jackson Laboratory | RRID: IMSR_JAX: 031331 | |

*Continued*

| Reagent type (species) or resource | Designation | Source or reference | Identifiers | Additional information |
|---|---|---|---|---|
| Strain, strain background (*M. musculus*) | B6J.129S6(FVB)-Slc32a1$^{tm2(cre)Lowl}$/MwarJ, Slc32a1-IRES-Cre | The Jackson Laboratory | RRID: IMSR_JAX:028862 | |
| Strain, strain background (*M. musculus*) | B6;129S6-Sncg$^{em1(flpo)Hze}$/J, Sncg-IRES2-FlpO-neo | The Jackson Laboratory | RRID: IMSR_JAX:034424 | |
| Strain, strain background (*M. musculus*) | B6J.Cg-Sst$^{tm2.1(cre)Zjh}$/MwarJ, Sst-IRES-Cre | The Jackson Laboratory | RRID: IMSR_JAX:028864 | |
| Strain, strain background (*M. musculus*) | B6;129S-Tac1$^{tm1.1(cre)Hze}$/J, Tac1-IRES2-Cre | The Jackson Laboratory | RRID: IMSR_JAX:021877 | |
| Strain, strain background (*M. musculus*) | B6.FVB(Cg)-Tg(Th-cre)FI172Gsat/Mmucd, Th-Cre_FI172 | MMRRC | RRID: MMRRC_031029-UCD | |
| Strain, strain background (*M. musculus*) | C57BL/6N-Th$^{tm1Awar}$/Mmmh, Th-P2A-FlpO or TH-2A-Flpo | MMRRC | RRID: MMRRC_050618-MU | |
| Strain, strain background (*M. musculus*) | B6.FVB(Cg)-Tg(Tlx3-cre)PL56Gsat/Mmucd, Tlx3-Cre_PL56 | MMRRC | RRID: MMRRC_041158-UCD | |
| Strain, strain background (*M. musculus*) | B6.FVB(Cg)-Tg(Tlx3-cre)PL56Gsat/Mmucd, Tlx3-Cre_PL56 | MMRRCC | RRID: MMRRC_041158-UCD | |
| Strain, strain background (*M. musculus*) | B6J.Cg-Vip$^{tm1(cre)Zjh}$/AreckJ, Vip-IRES-Cre | The Jackson Laboratory | RRID: IMSR_JAX:031628 | |
| Strain, strain background (*M. musculus*) | B6.Cg-Npy$^{tm1.1(flpo)Hze}$/J, Npy-IRES2-FlpO | The Jackson Laboratory | RRID: IMSR_JAX:030211 | |
| Strain, strain background (*M. musculus*) | B6;129S-*Vipr2*$^{tm1.1(cre)Hze}$/J, Vipr2-IRES2-Cre | The Jackson Laboratory | RRID: IMSR_JAX: 031332 | |
| Commercial assay or kit | SMART-Seq v4 Ultra Low Input RNA Kit for Sequencing | Takara | 634,894 | |
| Commercial assay or kit | Nextera XT Index Kit V2 Set A-d | Nextera | FC-131–2001, 2002, 2003, 2004 | |
| Software, algorithm | IGOR pro | Wavemetrics | RRID:SCR_000325 | |
| Software, algorithm | MIES: Software for Electrophysiology functions | Allen Institute | RRID:SCR_016443 | |
| Software, algorithm | Vaa3d | *Peng et al., 2010* | https://alleninstitute.org/what-we-do/brain-science/research/products-tools/vaa3d/ | |
| Software, algorithm | IPFX | Allen Institute | https://github.com/alleninstitute/ipfx | *Matt, 2021* |
| Software, algorithm | DRCME | Allen Institute, *Gouwens et al., 2019* | https://github.com/alleninstitute/drcme | *Nathan W, 2021* |

## Mouse breeding and husbandry

All procedures were carried out in accordance with the Institutional Animal Care and Use Committee at the Allen Institute for Brain Science. Animals (< 5 mice per cage) were provided food and water ad libitum and were maintained on a regular 12 hr light–dark cycle. Animals were maintained on the C57BL/6 J background, and newly received or generated transgenic lines were backcrossed to C57BL/6 J. Experimental animals were heterozygous for the recombinase transgenes and the reporter transgenes.

## Human tissue acquisition

Human tissue acquisition details can be found here (*Berg et al., 2020*). Briefly, surgical specimens were obtained from local hospitals (Harborview Medical Center, Swedish Medical Center and University of Washington Medical Center) in collaboration with local neurosurgeons. All patients provided informed consent and experimental procedures were approved by hospital institute review boards before commencing the study. Human surgical tissue specimens were immediately transported (15–35 min) from the hospital site to the laboratory for further processing.

## Tissue processing

For mouse experiments male and females were used between the ages of P45 and P70 and anesthetized with 5 % isoflurane and intracardially perfused with 25 mL of 0–4°C slicing ACSF.

Human, mouse or macaque slices (350 µm) were generated (Compresstome VF-300 vibrating microtome, Precisionary Instruments or VT1200S Vibratome, Leica Biosystems), with a block-face image acquired (Mako G125B PoE camera with custom integrated software) before each section to aid in registration to the common mouse reference atlas. Brains were mounted for slicing either coronally or 17° off-coronal to preserve intactness of neuronal processes in primary visual cortex.

Slices were transferred to an oxygenated and warmed (34 °C) slicing ACSF for 10 min, then transferred to room temperature holding ACSF (continuously bubbled with 95 % $O_2$/5 % $CO_2$) for the remainder of the day until transferred for patch-clamp recordings.

## Patch-clamp recording

Slices were bathed in warm (34 °C) recording ACSF and continuously bubbled with 95 % $O_2$/5 % $CO_2$. The bath solution contained blockers of fast glutamatergic (1 mM kynurenic acid) and GABAergic synaptic transmission (0.1 mM picrotoxin). Thick-walled borosilicate glass (Warner Instruments, G150F-3) electrodes were manufactured (Narishige PC-10) with a resistance of 4–5 MΩ. Before recording, the electrodes were filled with ~1.0–1.5 µL of internal solution with biocytin (110 mM potassium gluconate, 10.0 mM HEPES, 0.2 mM ethylene glycol-bis (2-aminoethylether)-N,N,N',N'-tetraacetic acid, 4 mM potassium chloride, 0.3 mM guanosine 5'-triphosphate sodium salt hydrate, 10 mM phosphocreatine disodium salt hydrate, 1 mM adenosine 5'-triphosphate magnesium salt, 20 µg/mL glycogen, 0.5 U/µL RNAse inhibitor (Takara, 2,313A) and 0.5 % biocytin (Sigma B4261), pH 7.3). The pipette was mounted on a Multiclamp 700B amplifier headstage (Molecular Devices) fixed to a micromanipulator (PatchStar, Scientifica).

The composition of bath and internal solution as well as preparation methods were made to maximize the tissue quality of slices from adult mice, to align with solution compositions typically used in the field (to maximize the chance of comparison to previous studies), modified to reduce RNAse activity and ensure maximal gain of mRNA content.

Electrophysiology signals were recorded using an ITC-18 Data Acquisition Interface (HEKA). Commands were generated, signals processed, and amplifier metadata were acquired using MIES written in Igor Pro (Wavemetrics). Data were filtered (Bessel) at 10 kHz and digitized at 50 kHz. Data were reported uncorrected for the measured (*Neher, 1992*)–14 mV liquid junction potential between the electrode and bath solutions.

Prior to data collection, all surfaces, equipment and materials were thoroughly cleaned in the following manner: a wipe down with DNA away (Thermo Scientific), RNAse Zap (Sigma-Aldrich), and finally nuclease-free water.

After formation of a stable seal and break-in, the resting membrane potential of the neuron was recorded (typically within the first minute). A bias current was injected, either manually or automatically using algorithms within the MIES data acquisition package, for the remainder of the experiment to maintain that initial resting membrane potential. Bias currents remained stable for a minimum of 1 s before each stimulus current injection.

To be included in analysis, a neuron needed to have a > 1 GΩ seal recorded before break-in and an initial access resistance <20 MΩ and <15 % of the $R_{input}$. To stay below this access resistance cut-off, neurons with a low input resistance were successfully targeted with larger electrodes. For an individual sweep to be included, the following criteria were applied: (1) the bridge balance was <20 MΩ and <15 % of the $R_{input}$; (2) bias (leak) current 0 ± 100 pA; and (3) root mean square noise

measurements in a short window (1.5 ms, to gauge high frequency noise) and longer window (500 ms, to measure patch instability) were <0.07 mV and 0.5 mV, respectively.

Cell health rating is a subjective call ranked on a scale of 1–5, with one corresponding to poor health and five corresponding to healthy. Scores were based on the unique visible features of neuron prior to patching. Cells ranked one included shriveled and/or dark edges or swollen appearances. Whereas a cell that ranked five included features such as a well-defined soma, soft edges, and a three-dimensional appearance.

Extracting the nucleus at the conclusion of the electrophysiology experiment led to a substantial increase in transcriptomic data quality. Upon completion of electrophysiological examination, the pipette was centered on the soma or placed near the nucleus (if visible). A small amount of negative pressure was applied (~–30 mbar) to begin cytosol extraction and attract the nucleus to the tip of the pipette. After approximately one minute, the soma had visibly shrunk and/or the nucleus was near the tip of the pipette. While maintaining the negative pressure, the pipette was slowly retracted in the x and z direction. Slow, continuous movement was maintained while monitoring pipette seal. Once the pipette seal reached >1 GΩ and the nucleus was visible on the tip of the pipette, the speed was increased to remove the pipette from the slice. The pipette containing internal solution, cytosol, and nucleus was removed from the pipette holder and contents were expelled into a PCR tube containing lysis buffer (Takara, 634894).

## Electrophysiology feature analysis

Electrophysiological responses were elicited by short (3 ms) current pulses, (1 s) current steps, and a slow ramp (25 pA/s). These stimulation protocols and the following feature extraction methods were previously described in *Gouwens et al., 2019* and (*Gouwens et al., 2020*). Briefly, AP locations were identified where the smoothed derivative of the membrane potential (dV/dt) exceeded 20 mV/ms, and were further refined based on threshold-to-peak voltage, time differences and absolute peak height. The AP threshold was adjusted to where the dV/dt was 5 % of the average maximal dV/dt across all APs. Features including peak, width (at half-height) and upstroke/downstroke ratio (i.e. ratio of the peak upstroke dV/dt to the peak downstroke dV/dt) were calculated for each AP.

As described in *Gouwens et al., 2019*, a sparse principal components analysis (sPCA) was applied to sets of feature vectors that included (a) concatenated waveforms of the first APs evoked by the lowest-amplitude current steps (b) the derivatives of these waveforms, (c) interspike intervals, between the fast trough of the initial AP and the threshold of the following AP, normalized in duration and averaged, (d) binned (in 20 ms intervals) features derived from long current steps at stimulation amplitudes at rheobase and +40 pA and +80 pA above rheobase, with interpolation from neighboring sweeps if a sweep at the expected amplitude was not available, and normalization of instantaneous firing frequency to the maximum value for the step, (e) downsampled (to 10 ms bins) and concatenated subthreshold responses from –10 pA to –90 pA steps (at a –40 pA interval), (f) response to the largest available hyperpolarizing step, normalized to the minimum and baseline membrane potentials. sPCA was performed separately for the feature vector data in each of the categories described above. PCs with an adjusted explained variance greater than 1 % were kept, yielding a total of 45 components, which were then z-scored, combined and visualized using Uniform Manifold Approximation and Projection (*Becht et al., 2018*).

## cDNA amplification and library construction

We used the SMART-Seq v4 Ultra Low Input RNA Kit for Sequencing (Takara, 634894) to reverse transcribe poly(A) RNA and amplify full-length cDNA according to the manufacturer's instructions. We performed reverse transcription and cDNA amplification for 20 PCR cycles in 0.65 ml tubes, in sets of 88 tubes at a time. At least one control eight-strip was used per amplification set, which contained four wells without cells and four wells with 10 pg control RNA. Control RNA was either Universal Human RNA (UHR) (Takara 636538) or control RNA provided in the SMART-Seq v4 kit. All samples proceeded through Nextera XTDNA Library Preparation (Illumina FC-131–1096) using either Nextera XT Index Kit V2 Sets A-D (FC-131–2001, 2002, 2003, 2004) or custom dual-indexes provided by IDT (Integrated DNA Technologies). Nextera XT DNA Library prep was performed according to manufacturer's instructions, except that the volumes of all reagents including cDNA input were decreased to 0.2 x by volume. Each sample was sequenced to approximately 500 k reads.

## RNA-sequencing

Fifty-base-pair paired-end reads were aligned to GRCm38 (mm10) using a RefSeq annotation gff file retrieved from NCBI on 18 January 2016 (https://www.ncbi.nlm.nih.gov/genome/annotation_euk/all). Sequence alignment was performed using STAR v2.5.3 (*Dobin et al., 2012*) in two pass Mode. PCR duplicates were masked and removed using STAR option 'bamRemoveDuplicates'. Only uniquely aligned reads were used for gene quantification. Gene counts were computed using the R Genomic Alignments package (*Lawrence et al., 2013*). Overlaps function using 'IntersectionNotEmpty' mode for exonic and intronic regions separately. Exonic and intronic reads were added together to calculate total gene counts; this was done for both the reference dissociated cell data set and the Patch-seq data set of this study.

## Mapping to VISp reference taxonomy

As described in *Gouwens et al., 2020*, Patch-seq data was mapped to the reference taxonomy prepared from dissociated cells from *Tasic et al., 2018*, using only neuronal cells and their corresponding cell types from VISp region. The mapping was done in a top down manner, first resolving subclasses then types. At each branch point, the correlation between the given transcriptome and the reference cell types was computed using the marker genes associated with that branch point. The most correlated branch was chosen and the process was repeated down to the leaves. Mapping confidence was determined with 100 bootstrapped iterations at each branch point, sampling a subset (70%) of reference cells and markers (70%), as described in *Gouwens et al., 2020*. The cell was assigned the cell type to which the cell had mapped with highest probability.

## Mapping quality assessment

We used the same methodology as in *Gouwens et al., 2020* to assess the quality of mapping and to quantify the expected versus unexpected ambiguity of mapping between cell types. First, the reference dissociated cells were mapped to the reference taxonomy, and a reference mapping probability matrix was constructed. The Kullback-Leibler (KL) divergence (*Berger et al., 2009*) between the mapping probability distributions of Patch-seq cells and the reference mapping probability distribution was computed. Cells with a KL divergence greater than two were categorized as 'inconsistent'. The correlation in gene expression for each remaining cell and the reference cells from the same cell type to which that cell was assigned was the computed. If the correlation was less than 0.5, the cell was also categorized as 'inconsistent'. Probability of mapping to one or more than one type was used to assign the quality of the remaining cells. A cell was categorized as 'highly consistent' if the sum of its two highest mapping probabilities exceeded 70 % and if the ratio of the highest mapping probability to the second highest was more than 2. Otherwise the cell was categorized as 'moderately consistent'.

## Biocytin histology

A horseradish peroxidase (HRP) enzyme reaction using diaminobenzidine (DAB) as the chromogen was used to visualize the filled neurons after electrophysiological recording, and 4,6-diamidino-2-phenylindole (DAPI) stain was used identify cortical layers.

## Imaging

Mounted sections were imaged as described previously (*Gouwens et al., 2019*). Briefly, operators captured images on an upright AxioImager Z2 microscope (Zeiss, Germany) equipped with an Axiocam 506 monochrome camera and 0.63 x optivar. Two-dimensional tiled overview images were captured with a 20 X objective lens (Zeiss Plan-NEOFLUAR 20 X/0.5) in brightfield transmission and fluorescence channels. Tiled image stacks of individual neurons were acquired at higher resolution in the transmission channel only for the purpose of automated and manual reconstruction. Light was transmitted using an oil-immersion condenser (1.4 NA). High-resolution stacks were captured with a 63 X objective lens (Zeiss Plan-Apochromat 63 x/1.4 Oil or Zeiss LD LCI Plan-Apochromat 63 x/1.2 Imm Corr) at an interval of 0.28 µm (1.4 NA objective) or 0.44 µm (1.2 NA objective) along the Z axis. Tiled images were stitched in ZEN software and exported as single-plane TIFF files.

## Anatomical location

To characterize the position of biocytin-labeled neurons in the mouse brain, a 20 x brightfield and/or fluorescent image of DAPI-stained tissue was captured and analyzed to determine layer position and

region. Soma position of reconstructed neurons was annotated and used to calculate soma depth relative to drawings of the pia and/or white matter. Individual mouse neurons were also manually placed in the appropriate cortical region and layer within the Allen Mouse Common Coordinate Framework (CCF) by matching the 20 x image of the slice with a 'virtual' slice at an appropriate location and orientation within the CCF. Laminar locations were calculated by finding the path connecting pia and white matter that passed through the neuron's coordinate, identifying its distance to pia and white matter as well as position within its layer, then aligning those values to an average set of layer thicknesses. Using the DAPI image, laminar borders were also drawn for all reconstructed neurons.

## Morphological reconstruction

Reconstructions were generated based on a 3D image stack that was run through a Vaa3D-based image processing and reconstruction pipeline (*Peng et al., 2010*). Initial reconstructions were generated with an automated reconstruction of the neuron using TReMAP (*Zhou et al., 2015*), using reconstruction software PyKNOSSOS (Ariadne-service) or the citizen neuroscience game Mozak (Mozak. science) (*Roskams and Popović, 2016*). Automated or manually-initiated reconstructions were then extensively manually corrected and curated using a range of tools (e.g. virtual finger, polyline) in the Mozak extension (Zoran Popovic, Center for Game Science, University of Washington) of Terafly tools (*Bria et al., 2016*; *Peng et al., 2014*) in Vaa3D. Every attempt was made to generate a completely connected neuronal structure while remaining faithful to image data. If axonal processes could not be traced back to the main structure of the neuron, they were left unconnected.

Before morphological feature analysis, reconstructed neuronal morphologies were expanded in the dimension perpendicular to the cut surface to correct for shrinkage (*Egger et al., 2007*; *Deitcher et al., 2017*) after tissue processing. The amount of shrinkage was calculated by comparing the distance of the soma to the cut surface during recording and after fixation and reconstruction. A tilt angle correction was also performed based on the estimated difference (via CCF registration) between the slicing angle and the direct pia-white matter direction at the neuron's location (*Gouwens et al., 2019*).

## Morphological quality assessments

All slices received a 20 x single image scan to evaluate the biocytin fill of patched neurons. If the soma failed to retain the biocytin and was not visible, then the morphological assessment was categorized as 'failed'. To be strategic about which cells are imaged, high-resolution (63 x) images were collected only for Patch-seq experiments with passing cDNA (subjective call), passing NMS (> 0.4) and a visible neuron in the 20 x scan. If the neuron was not determined to be suitable for digital reconstruction from 63 x stack collection, it was categorized as 'medium quality'. Neurons that were of the highest quality and suitable for digital reconstructions were categorized as 'high quality'. Upon detailed inspection, some neurons originally destined for high-resolution stack collection and reconstruction were determined to have an 'insufficient axon'. Neurons that had an observable fill on the 20 x image but failed either cDNA assessment or NMS were classified as 'not assessed'.

## Acknowledgements

We thank the following teams for the services and support. Reagent prep and immunohistochemistry: Medea McGraw, Kris Bickley, Jasmine Bomben, Krissy Brouner, Tom Egdorf, Amanda Gary, Michelle Maxwell, Daniel Park, Alice Pom, and Augustin Ruiz. Human and Mouse Tissue Processing: Nick Dee, Elizabeth Barkan, Tamara Casper, Kristen Crichton, Matt Kroll, Josef Sulc, and Herman Tung. Macaque Tissue Processing: Jonathan Ting, Victoria Omstead, and Natalie Weed. Molecular biology: Darren Bertagnolli, Jeff Goldy, Delissa McMillen, and Michael Tieu. Transcriptomics: Fahimeh Baftizadeh. Imaging: Rusty Nicovich, Rachel Enstrom, Melissa Gorham, Maddie Hupp, Samuel Lee, and Lydia Potekhina. Morphology and Reconstructions: Rusty Nicovich, Lauren Alfiler, Alex Henry, Sara Kebede, Matt Mallory, Alice Mukora, David Sandman, Grace Williams and the Mozak citizen-scientists. We are thankful for our collaborator neurosurgeons at the local hospital sites: Charles Cobbs, Richard Ellenbogen, Manuel Ferreira, Ryder Gwinn, Andrew Ko, Jeffrey Ojemann, Akshal Patel, Daniel Silbergeld. We appreciate feedback on the manuscript provided by Shreejoy Tripathy and Xiao Luo. Funding: NIH grants P51OD010425 (BEK) from the Office of Research Infrastructure Programs (ORIP), UL1TR000423 (BEK) from the National Center for Advancing Translational Sciences (NCATS), and U01 MH114812-02

(ESL). This work was funded by the Allen Institute for Brain Science. We wish to thank the Allen Institute founder, Paul G Allen, for his vision, encouragement, and support.

## Additional information

### Funding

| Funder | Grant reference number | Author |
|---|---|---|
| NIH Office of the Director | P51OD010425 | Brian E Kalmbach Jonathan T Ting |
| National Center for Advancing Translational Sciences | UL1TR000423 | Brian E Kalmbach Jonathan T Ting |
| National Institute of Mental Health | U01 MH114812-02 | Ed Lein |

The funders had no role in study design, data collection and interpretation, or the decision to submit the work for publication.

### Author contributions

Brian R Lee, Conceptualization, Data curation, Formal analysis, Investigation, Methodology, Project administration, Supervision, Validation, Visualization, Writing – original draft, Writing – review and editing; Agata Budzillo, Jim Berg, Conceptualization, Data curation, Formal analysis, Investigation, Methodology, Project administration, Resources, Software, Supervision, Validation, Visualization, Writing – original draft, Writing – review and editing; Kristen Hadley, Katherine Baker, Data curation, Investigation, Methodology, Validation, Writing – review and editing; Jeremy A Miller, Formal analysis, Investigation, Resources, Validation, Visualization, Writing – original draft, Writing – review and editing; Tim Jarsky, Formal analysis, Investigation, Resources, Software, Validation, Writing – review and editing; DiJon Hill, Sara Vargas, Data curation; Lisa Kim, Lindsay Ng, Aaron Oldre, Jessica Trinh, Data curation, Investigation, Methodology, Validation; Rusty Mann, Data curation, Formal analysis, Investigation, Methodology, Resources, Validation, Writing – review and editing; Ram Rajanbabu, Data curation, Investigation, Validation; Thomas Braun, Resources, Software, Validation; Rachel A Dalley, Formal analysis, Investigation, Methodology, Validation, Visualization, Writing – review and editing; Nathan W Gouwens, Investigation, Methodology, Resources, Software, Validation, Visualization; Brian E Kalmbach, Data curation, Funding acquisition, Investigation, Methodology, Validation, Visualization, Writing – review and editing; Tae Kyung Kim, Conceptualization, Data curation, Investigation, Methodology; Kimberly A Smith, Data curation, Investigation, Resources, Supervision, Validation; Gilberto Soler-Llavina, Conceptualization, Investigation, Methodology, Project administration, Supervision, Visualization, Writing – review and editing; Staci Sorensen, Hongkui Zeng, Conceptualization, Investigation, Methodology, Project administration, Supervision, Validation, Visualization; Bosiljka Tasic, Conceptualization, Data curation, Investigation, Methodology, Project administration, Supervision, Validation, Visualization; Jonathan T Ting, Conceptualization, Data curation, Funding acquisition, Investigation, Methodology, Validation, Visualization, Writing – review and editing; Ed Lein, Conceptualization, Funding acquisition, Investigation, Methodology, Project administration; Gabe J Murphy, Conceptualization, Investigation, Methodology, Project administration, Supervision

### Author ORCIDs

Brian R Lee ⓘ http://orcid.org/0000-0002-3210-5638
Agata Budzillo ⓘ http://orcid.org/0000-0002-2723-3272
Jeremy A Miller ⓘ http://orcid.org/0000-0003-4549-588X
Tim Jarsky ⓘ http://orcid.org/0000-0002-4399-539X
Rusty Mann ⓘ http://orcid.org/0000-0002-0226-2069
Thomas Braun ⓘ http://orcid.org/0000-0002-1416-2065
Nathan W Gouwens ⓘ http://orcid.org/0000-0001-8429-4090
Bosiljka Tasic ⓘ http://orcid.org/0000-0002-6861-4506
Hongkui Zeng ⓘ http://orcid.org/0000-0002-0326-5878

Jim Berg  http://orcid.org/0000-0002-3300-5399

### Ethics

Human subjects: De-identified human brain tissue and data used in this research was collected by local hospitals during clinically necessary surgery. Study participants gave informed consent to share their de-identified tissue and data either with the Allen Institute specifically or more broadly with collaborators of the study PIs prior to surgery. Participants consented to share their de-identified genomic data in controlled access in compliance with National Institutes of Health Genomic Data Sharing policy. The study participants were informed that the resulting data might be broadly shared, through publications, presentations, or scientific repositories and of the potential risks of sharing these data. Samples obtained from the Swedish Neuroscience Institute were collected under approved Western Institutional Review Board protocols (#1111798 and #1068035) in collaboration with Drs. Charles Cobb and Ryder Gwinn respectively. Samples obtained from Harborview Medical Center were obtained under approval of the University of Washington Institutional Review Board protocol (#HSD No. 49119) in collaboration with Dr. Jeffrey Ojemann.

The animal research in this study was performed in accordance with the Guide for the Care and Use of Laboratory Animals and the Public Health Service Policy on Humane Care and Use of Laboratory Animals in compliance with National Institutes of Health policy. All housing, handling, and experimental use of the animals occurred with the oversight and approval of the Allen Institute Institutional Animal Care and Use Committee (Protocol 1809). All surgeries and retro-orbital injections were performed under isoflurane anesthesia with perioperative analgesics and fluid support.

### Decision letter and Author response

Decision letter https://doi.org/10.7554/eLife.65482.sa1
Author response https://doi.org/10.7554/eLife.65482.sa2

---

## Additional files

### Supplementary files

• Supplementary file 1. Patch-seq workflow states and QC definitions. The purpose of this table is to list the different parameters and purpose for each stage within the Patch-seq protocols, the metrics measured, the subsequent data and range of data obtained. The criteria that the Allen Institute uses for Patch-seq can be adopted, relaxed, or disregarded. [1] Electrophysiology, morphology and transcriptomic parameters and metrics can also be found in the technical white papers at http://celltypes.brain-map.org.

• Transparent reporting form

• Source data 1. Human MET metadata.

### Data availability

The data used in this manuscript, the software packages, the detailed protocol, and online resources are freely available to the public and have been consolidated at https://github.com/AllenInstitute/patchseqtools (copy archived at https://archive.softwareheritage.org/swh:1:rev:d1afcd4d5203564979a29f2891e03cba7733b726).

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
