## [Decision Letter]

**Acceptance summary:**

The authors provide detailed data and protocols useful for optimizing the patch-seq technique for simultaneously characterizing the morphology, electrophysiology and gene expression of single neurons. The refined protocol is likely to be a useful resource for ongoing efforts to map neuronal cell type taxonomy.

**Decision letter after peer review:**

Thank you for submitting your article "Scaled, high fidelity electrophysiological, morphological, and transcriptomic cell characterization" for consideration by *eLife*. Your article has been reviewed by 3 peer reviewers, including Sacha B Nelson as the Reviewing Editor and Reviewer #1, and the evaluation has been overseen by Lu Chen as the Senior Editor. The following individual involved in review of your submission has agreed to reveal their identity: Chris J McBain (Reviewer #3).

The reviewers have discussed their reviews with one another, and the Reviewing Editor has drafted this to help you prepare a revised submission. Overall, the reviewers recognize the significance of your study, which provides a detailed protocol for obtaining high quality Patch-seq data, with all the resources available to the community. The reviewers thus recommended this submission be published as a Tools and Resources piece upon successful revision.

Summary:

The reviewers and editors were divided in their assessment of your manuscript. All saw merit in the thoroughness of the approach and the importance of fully describing the method to those seeking to emulate it. However, what was difficult to assess is precisely what is new here. The suitability for publication in eLife will depend on clarifying the data, findings and protocols that have not been previously reported.

Essential revisions:

1) The authors should be entirely transparent within the paper about which (if any) data are being reported for the first time and which have been previously described. The same is true for the resources provided (e.g. the software) and for the methodological conclusions. It is fine to include material/ideas etc that were previously reported but these must be clearly indicated with phrases like "as previously reported." In general it is important to make the case that new details are provided which significantly improve the ability of others to carry out the technique, over and above the material already provided as part of publication of the other papers in this series.

2) The authors should comment on the similarities and differences between the current protocol and others recently published.

3) The issue of the granularity of cell type identification should be quantitatively addressed. Was it possible to identify all cells included in terms of the finest divisions identified in prior RNAseq studies including those in this current series? To the extent that this was not possible the reasons should be addressed. There are valid reasons to expect that the reduced sampling provided by patch seq may not be able to match the granularity of whole cell studies, but this is an important issue to address in detail, given the more complete scope of the data provided.

4) Criteria used to make classifications (e.g. such as cell health) should be specified.

*Reviewer #1:*

This is one of a series of recent papers from the Allen Institute using patch seq to characterize gene expression, electrophysiology and morphology of cortical neurons. The main focus of the present manuscript is to communicate improvements in the method developed in the course of acquiring large datasets that are more completely described in other papers including Gouwens et al. 2019, 2020 and Berg 2020.

What is difficult to assess is the relationship of the datasets and conclusions described here to the datasets and conclusions in these other papers. This should be communicated more clearly and transparently. Many of the points and resources presented in this paper are also presented in these other papers. For example, the availability of the electrophysiology acquisition software is noted in the 2019 Nature Neuroscience paper and the recent Cell paper and the importance of retaining the nucleus for improving transcriptome representation is described in this paper and in the Berg manuscript.

The major results that appear to be new here are:

1) A description of the electrode resistance during automated nucleus retrieval and extraction (Figure 2).

2) Perhaps, a more complete description of the automated electrophysiology software and protocol used in the recent papers (Figure 3), although this is described in some detail in the methods of Gouwens et al. 2020. If there is something new here it should be distinguished more clearly.

3) quantification of the effects of nucleus retrieval on marker expression (Figure 4)

And

4) the effects of nucleated patch resistance and recording duration on recovery of fills. (Figure. 5)

Figure 1 is a diagram and Figure 6 is a single example slice, presumably from the dataset in the Berg manuscript (but again this is hard to assess).

One important difference between the present transcriptomic/morphoelectric data and that presented in Gouwens et al. 2020, is that here there is no effort made to subdivide the Pvalb, Vip and SST subtypes of interneurons into the 28 subtypes identified in the Gouwens et al. paper. If these are an overlapping set of cells they should be organized in the same way. If instead these are a separate dataset, do they support the subtypes seen in the other study?

*Reviewer #2:*

Simultaneous, triple-modal profiling of an individual neuron (morphology, electrophysiology, and transcriptome) using the Patch-seq is an important approach to better understand each neuron and improve their classification, but is difficult to achieve. To improve this approach, Lee and his colleagues made several modifications to the Patch-seq protocol, aiming at increasing the efficiency of each step to minimize data attrition and enhance the throughput to facilitate a more comprehensive cell type characterization. These modifications include adopting a standard automatic electrophysiology software equipped with an online analysis function, nucleated patch to increase RNA yield, and optimized variables for the morphology recovery.

The manuscript was well written with enough technical details for anyone interested to follow. In general, the conclusions and technical claims were well supported by impressive quantitative analysis, and the data and figures are high-quality and impressive. There is no doubt that this refined protocol could be a useful technical resource for the field, especially for current cell type taxonomy efforts. However, given that a similar Patch-seq approach ("nucleated" patch, or 'cork' of the nucleus) has been recently published (Lipovsek 2020), I would not consider this protocol a significant technical advancement in the field. In my opinion, it is more appropriate to publish this as a protocol paper in those journals focusing on the protocol and methods (Star Protocol and Nature Protocol et al), other than an article in *eLife*. In addition, there were a few claims that could be wrong if their data could be analyzed differently (see Suggestions).

One major concern is the success rate of morphology recovery in the Patch-seq protocol. The authors claim the patch recording time has no impact on morphology quality. Based on our experience, this may be true for excitatory neurons, but not for inhibitory neurons. The axon of excitatory neurons is inevitably cut in slice preparation and their dendrite recovery is easy to achieve with minimal recording time. The real bottleneck here is labeling the axonal tree of the inhibitory neurons, which are generally much more intact and complex. The longer the patch recording is, the more likely biocytin diffuses and reaches into the fine tip of complex and highly-arborized axonal trees. So high-quality morphology recovery of inhibitory neurons critically depends on the recording time. I would suggest authors regroup their data based on cell type and assess the relationship between recording time and morphology quality by cell type, or even by subtype. After parsing this relationship by cell type, I believe we will see the significant impact of recording time variable on the morphology quality, particularly on interneurons.

Another comment is about how to assess cell health. It does not make sense to rank cell health in slice electrophysiology on a scale of 1 to 5. This is not practical and could run into being very subjective (for instance, what is the difference between 4 and 5, which factor determines this cell is 4 other than 5?). I would recommend a scale of 1 to 3.

*Reviewer #3:*

I think this is a tremendous paper, which will be extremely useful to everyone interested in combining cell physiology (electrophysiology and morphology) with gene expression profiling approaches. The authors do a great job at laying out how to best approach this problem and facilitate this by the provision of a suite of software that will allow combining electrophysiology with gene analysis.

I don't have any major issues with the manuscript and think it could pretty much be published as is.

---

## [Author Response]

Essential revisions:1) The authors should be entirely transparent within the paper about which (if any) data are being reported for the first time and which have been previously described. The same is true for the resources provided (e.g. the software) and for the methodological conclusions. It is fine to include material/ideas etc that were previously reported but these must be clearly indicated with phrases like "as previously reported." In general it is important to make the case that new details are provided which significantly improve the ability of others to carry out the technique, over and above the material already provided as part of publication of the other papers in this series.

Throughout the manuscript we have clarified where data have been previously described. We have added a column to the cell database to indicate if that cell has been included in a previous study focused on morpho-electric-transcriptomic cell types in the mouse visual cortex or human temporal cortex. We added a first paragraph to the Results section that clarifies the overlap in cells that are included in this manuscript versus the previous studies, including a Venn diagram as a supplement to figure 1.

We have also more clearly clarified how the techniques presented here differ from those mentioned in the methods portion of previous manuscripts. Specifically, we highlighted the data that drove the decision to pursue nucleated patches as well as the analysis functions built on top of the MIES electrophysiology system that allows for fast, high quality data acquisition.

2) The authors should comment on the similarities and differences between the current protocol and others recently published.

We have added commentary on recently published Patch-seq protocols and how the approach we present is similar and different.

3) The issue of the granularity of cell type identification should be quantitatively addressed. Was it possible to identify all cells included in terms of the finest divisions identified in prior RNAseq studies including those in this current series? To the extent that this was not possible the reasons should be addressed. There are valid reasons to expect that the reduced sampling provided by patch seq may not be able to match the granularity of whole cell studies, but this is an important issue to address in detail, given the more complete scope of the data provided.

We have added a new figure that addresses the mapping success of Patch-seq cells, and how that mapping success relates to objective measures of transcriptomic quality.

Originally, we avoided including mapping as a metric in this study in the interest of presenting a protocol that is independent of the transcriptomic landscape of the target region. However, the point that we can use the mapping quality as a measure of the objective transcriptomic quality metrics is valid, and the manuscript is much improved with this addition.

The new panels in figure 4 show the relationship between an objective measure of transcriptomic data quality, the normalized marker sum (NMS) and a measure of mapping quality used in Gouwens et al., 2020 A new supplement to figure 4 shows a side-by-side comparison of the mapping of Patch-seq (Gouwens et al., 2020) vs FACS data (Tasic et al., 2019).

4) Criteria used to make classifications (e.g. such as cell health) should be specified.

We have added to the methods section to describe the subjective criteria used to classify the health of cells. Additionally, we have included representative images of the different cell health scores in Figure 6—figure supplement 2.